



# Conditional diffusion models for downscaling & bias correction of Earth system model precipitation

\*

Michael Aich[1*], Philipp Hess[1,2], Baoxiang Pan[3], Sebastian Bathiany[1,2],
Yu Huang[1], Niklas Boers[1,2,4]

[1]Technical University Munich, Munich, Germany; School of Engineering & Design, Earth System Modelling.
[2]Potsdam Institute for Climate Impact Research, Potsdam, Germany.
[3]Institute of Atmospheric Physics, Chinese Academy of Sciences, Beijing, China.
[4]Global Systems Institute and Department of Mathematics, University of Exeter, Exeter, UK.

\*Corresponding author. E-mail: michael.aich@tum.de;

**Abstract**

Climate change exacerbates extreme weather events like heavy rainfall and flooding. As these events cause severe socioeconomic damages, accurate high-resolution simulation of precipitation is imperative. However, existing Earth System Models (ESMs) struggle resolving small-scale dynamics and suffer from biases. Traditional statistical bias correction and downscaling methods fall short in improving spatial structure, while recent deep learning methods lack controllability and suffer from unstable training. Here, we propose a machine learning framework for simultaneous bias correction and downscaling. We train a generative diffusion model purely on observational data. We map observational and ESM data to a shared embedding space, where both are unbiased towards each other and train a conditional diffusion model to reverse the mapping. Our method can correct any ESM field, as the training is independent of the ESM. Our approach ensures statistical fidelity, preserves large-scale spatial patterns and outperforms existing methods especially regarding extreme events.

**Short Summary:** Accurately simulating rainfall is essential to understand the impacts of climate change, especially extreme events such as floods and droughts. Climate models simulate the atmosphere at a coarse resolution and often misrepresent precipitation, leading to biased and overly smooth fields. We improve climate model precipitation using a generative machine learning model that is data-efficient, preserves key climate signals such as trends and variability, and significantly improves the representation of extreme events.





# 1 Introduction

With global warming, we anticipate more intense rainfall events and associated natural hazards, e.g., in terms of floods and landslides, in many regions of the world Lee et al. (2023). Understanding and accurately simulating precipitation is particularly important for adaptation planning and, hence, for mitigating damages and reducing risks associated with climate change. Earth System Models (ESMs) play a crucial role in simulating precipitation patterns for both historical and future scenarios. However, these simulations are computationally extremely demanding, primarily because they require solving complex partial differential equations. To manage the computational load, ESMs resort to approximate solutions on discretized grids with coarse spatial resolution (typically around 100 km). The consequence is that these models do not resolve small-scale dynamics, such as many of the processes relevant to precipitation generation. This leads to considerable biases in the ESM fields compared to observations. Moreover, the coarse spatial resolution prevents accurate projections of localized precipitation extremes. The precipitation fields simulated by ESMs can therefore not be used directly for impact assessments Zelinka et al. (2020) and especially tasks such as water resource and flood management, which require precise spatial data at high resolution Gutmann et al. (2014).

Statistical bias correction methods can be used as a post-processing to adjust statistical biases. Quantile mapping (QM) is the most common method for improving the statistics of ESM precipitation fields Tong et al. (2021); Gudmundsson et al. (2012); Cannon et al. (2015). QM reduces the bias using a mapping that, locally at each grid cell, aligns the estimated cumulative distribution of the model output with the observed precipitation patterns over a reference time period. Although QM is effective in correcting distributions of single grid cells, it falls short in improving the spatial structure and patterns of precipitation simulations Hess et al. (2022). Visual inspection shows that ESM precipitation remains too smooth compared to the observational data after applying quantile mapping.

To address these problems, deep learning methods have recently been introduced Hess et al. (2023); Pan et al. (2021); François et al. (2021); Hess et al. (2022). In these approaches, the statistical relationships between model simulations and observational data are learned implicitly. A general constraint when using machine learning methods for bias correction is that individual samples of observational and Earth System Model data are always unpaired. In this context, a sample is a specific weather situation at a specific point in time. The reason for this lack of pairs is that simulations, even with very similar initial conditions, diverge already after a short period of time due to the chaotic nature of the underlying atmospheric dynamics. Currently, one can, therefore, not utilize the wide range of supervised ML techniques that have shown great success in various disciplines in recent years and the available options are consequently restricted to self- and unsupervised machine learning methods. Recent studies Hess et al. (2023); Pan et al. (2021); François et al. (2021); Hess et al. (2022) applied generative adversarial networks (GANs Goodfellow et al. (2020)) and specifically cycleGANs Zhu et al. (2017) to improve upon existing bias correction techniques. A major limitation of GAN-based approaches is that the stability and convergence of the training process are difficult to control and that it is challenging to find metrics that indicate training convergence. In addition, GANs often suffer from mode collapse, where only a part of the target probability distribution is approximated by the GAN.

As noted above, the low spatial resolution of ESM fields prevents local risk and impact assessment, necessitating the additional use of downscaling methods. In line with the climate literature, we refer to increasing the spatial resolution as downscaling throughout our manuscript, although we are aware that, especially in the machine learning literature, the term upsampling is more prevalent. We use the term downscaling only when we want to increase the information content in an image as well as the number of pixels. When we refer to upsampling (downsampling), we only mean an increase (decrease) in the number of pixels. Statistical downscaling aims to learn a transformation from the low-resolution ESM fields to high-resolution observations. Recent developments lean towards using machine learning methods for this task Rampal et al. (2022); Hobeichi et al. (2023); Rampal et al. (2024). The potential for machine learning-based downscaling methods was already shown in Vandal et al. (2017); van der Meer et al. (2023); Doury et al. (2023, 2024); Rampal et al. (2025). The work presented a downscaling approach built on purely convolutional networks.

Recently, Hess et al. Hess et al. (2025) used an unconditional consistency model (CM) for downscaling 3° x 3.75° precipitation data to 0.75° x 0.9375°. However their target resolution is not sufficient for impact assessments. Our method improves upon the target resolution by downscaling to 0.25°.





We show that the consistency model applied to our higher resolution setting with limited amounts of training data struggles to approximate the distribution, highlighting an advantage of our conditional training approach. The analysis is further extended to out-of-distribution scenarios, particularly those involving extreme precipitation and future emission projections.

Diffusion models (DMs) have recently emerged as the state-of-the-art ML approach for conditional image generation Saharia et al. (2022b); Rombach et al. (2022); Saharia et al. (2022c) and image-to-image translation Saharia et al. (2022a), mostly outperforming GANs across different tasks. Diffusion models (Fig. 1 and fig. S1) avoid the common issues present with GANs in exchange for slower inference speed. A diffusion model consists of a forward and a backward process. During the forward process, noise is added to an image in subsequent steps to gradually remove its content. The amount of noise added follows a predefined equation. During the backward process, a neural network is trained to reverse each of these individual noising steps to recover the original image. The trained diffusion model can generate an image of the training data distribution, given a noise image as input. Recent work Wan et al. (2024) introduced a framework for downscaling and bias correction, combining a diffusion model that is responsible for downscaling and a model based on optimal transport responsible for bias correction Cuturi (2013). Optimal transport learns a map between two data distributions in an unsupervised setting. However, this framework is computationally expensive and has so far only been demonstrated on synthetic datasets, without evaluation on real-world observational or ESM fields. In contrast, our approach is computationally efficient by integrating computationally efficient QM for large-scale bias correction with a conditional diffusion model that performs both small-scale bias correction and downscaling by generating matching small-scale patterns. Our conditional design ensures that essential ESM properties, such as trends, uncertainties, and time consistency, are preserved. We demonstrate its effectiveness for precipitation data, highlighting its ability to correct biases, downscale accurately, and capture extremes, uncertainties, and trends. A major advantage is that our conditional training allows us to use a relatively small dataset for training and still capture the distribution accurately. In contrast, unconditional models often need considerably more data to capture the full data variability, as we also show in our comparison to Hess et al. (2025) (see fig. S20).

Existing work leveraging state-of-the-art ML methods for bias correction and downscaling does not systematically investigate out-of-distribution scenarios like future emission scenarios and especially the representation of extreme events of the generative models in detail. Understanding the generalization performance of the models under these conditions is, however, crucial for impact modelers who rely on these outputs for risk assessments under future climate conditions. We will therefore present a detailed analysis of the generalization capability of our approach, both in terms of its performance in preserving climate change trends, as well as in capturing extreme events and their trends.

A major challenge in bias correction and downscaling of ESMs is that the whole class of state-of-the-art supervised machine learning methods cannot be used to solve this task. This significantly restricts the applicability and therefore progress of most AI based methods for the task. The problem is especially hard because there are no pairs between high-resolution observation-like data and low-resolution biased ESM data. Our approach offers a crucial advantage by reformulating the problem in a novel conditional framework that circumvents the need for explicit ESM-observation-like pairs, thereby allowing the effective utilization of state-of-the-art supervised and self-supervised training. Our proposed method strikes a balance by leveraging the strengths of ESMs – such as temporal consistency, trends, and responses to external forcing – and integrating them with a novel generative approach for bias correction and downscaling.

We present here a framework based on state-of-the-art conditional diffusion models that allows us to perform both bias correction and downscaling with one single network. This network can, in principle, be any supervised machine learning method. We use a single conditional diffusion model (Fig. 1 and fig. S1) to correct low-resolution (LR) ESM fields toward high-resolution (HR) observational data (OBS). The supervised formulation of the task allows us to train a conditional diffusion model that is more data efficient than its unconditional counterpart because it is trained to only learn the small-scale precipitation patters, given the large-scale patters. An unconditional model that learns to approximate the full distribution of precipitation at all scales is unnecessarily complex for the task. In general, our task of bias correction and downscaling can be seen as taking a field from a distribution $p(ESM)$ and transforming it into a field from a conditional distribution $p(OBS|ESM)$.

We cannot learn such a downscaling and bias correction transformation directly, because ESM and OBS are unpaired. The key idea of our method is to generate OBS-like fields given perturbed versions of those fields in a conditional and paired way. At inference, we construct a perturbed version of the ESM data and ensure its statistical similarity to the perturbed OBS data. This allows our model to



generalize to ESM field conditions, while being trained only on OBS data. Our framework therefore
introduces transformations $f$ and $g$ that map ESM and OBS data to a shared embedding space (see
Methods and Fig. 1A). The embedding transformations match the statistics of the embedded data
$f(OBS)$ and $g(ESM)$ without creating pairs between the individual OBS and ESM fields. On this
shared embedding space, we can train a conditional diffusion model to approximate the inverse of $f$,
only relying on the pairs OBS and $f(OBS)$. We are then able to bias correct and downscale ESM
fields by first mapping them to the shared embedding space and then projecting them to the OBS
dataset with the trained diffusion model (Fig. 1B and Fig. 1C). The purpose of the embedding space is
to facilitate generalization between $f(OBS)$ and $g(ESM)$, allowing the diffusion model to be trained
without direct access to the ESM fields, while it can still be evaluated on $f(OBS)$. This framework
offers great flexibility as it can be applied to any ESM. The embedding transformation for the ESM
has two key components. First, we use quantile mapping (QM) as a fast and effective method to
correct large-scale biases in the ESM. Second, we introduce noise to remove small-scale information
in the precipitation fields. We define large scales as those spatial scales that are effectively corrected
using QM alone, while smaller spatial scales, which require additional correction, are referred to as
small scales (Fig. 2). This noise selectively targets small-scale patterns, leaving intact large-scale
patterns. In our approach, quantile mapping addresses large-scale biases, while the small-scale biases
and downscaling are handled by our diffusion model. The task of our model is then to perform
downscaling and bias correction by regenerating these small-scale features, in a way that ensures
consistency with the preserved large-scale patterns.




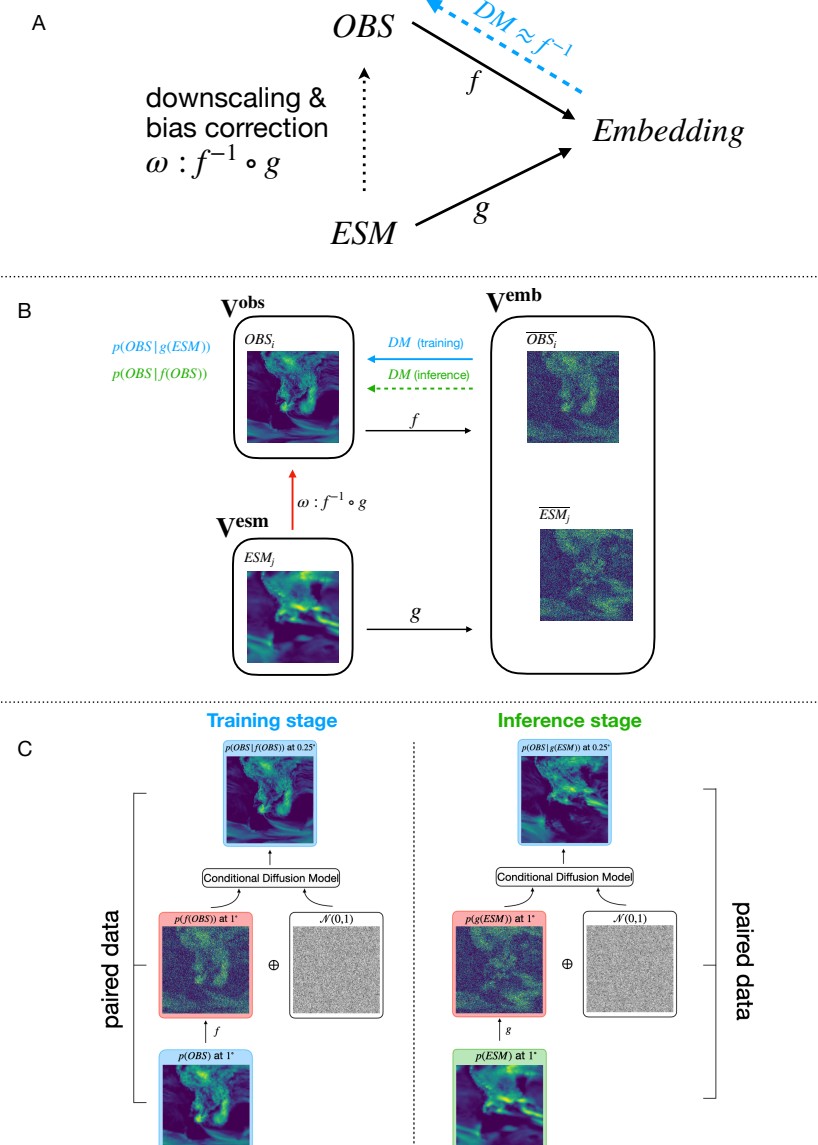

**Fig. 1  Schematic overview of our approach**. **(A)** Bias correction and downscaling can be formulated as a mapping $\omega$ from the ESM data space to the data space of observations (OBS) used for training. We first map both datasets to a shared embedding space and then learn the inverse of the mapping $f$ with a DM. We achieve a correction of the ESM data by applying $DM \circ g$. **(B)** Our framework allows to train a single model for bias correction and downscaling in a supervised way despite the unpaired nature of OBS and ESM fields. We construct functions $f, g$ that map $OBS \in \mathbf{V^{obs}}$ and $ESM \in \mathbf{V^{esm}}$ fields to a shared embedding space $\mathbf{V^{emb}}$. Note that this embedding space does not enforce pairing between individual fields, but a similar distribution between the embedded fields. By inverting $f$, we can rewrite $\omega$ as $\omega = f^{-1} \circ g$. We learn the inverse $f^{-1}$ with a conditional diffusion model. This model is trained (blue arrow) on pairs of observational data to approximate the map from $f(OBS)$ to OBS. Because $f(OBS)$ and $g(ESM)$ share the embedding space (and are identically distributed by construction), we can evaluate (green arrow) the $DM$ on the embedded ESM data $g(ESM)$ and thereby approximate the bias correction and downscaling function $\omega = f^{-1} \circ g \approx DM \circ g$, without the need of paired data between OBS and ESM. The indices $i, j$ highlight that the two exemplary fields $ESM_j$ and $ESM_i$ are not paired.
**(C)** Left: Training process of the conditional DM $DM \approx f^{-1}$. Note that the individual samples of the input OBS and their embeddings $f(OBS)$, as well as the embeddings $f(OBS)$ and the output of $DM \approx f^{-1}$ are paired, respectively. Right: Inference process of $DM \approx f^{-1}$. In this case, the individual samples of the input ESM, their embeddings $g(ESM)$, and the output of $DM \approx f^{-1}$ are paired, respectively. It is not necessary for the training embedding samples to be paired with the inference embedding samples. See fig. S1 for details.



## 2 Results

The ability of the diffusion model $DM$ to approximate $f^{-1}$ and the effectiveness of the transformations $f, g$ will determine the overall performance of the downscaling and bias correction model $\omega = DM \circ g$. Therefore, we first investigate the effectiveness of the embedding transformations $f$ and $g$, followed by an analysis of the downscaling and bias correction performance of the diffusion model $DM$, on the observational dataset. Once we have shown that both work as expected, we investigate the performance of the diffusion model in bias correction and downscaling of the ESM precipitation fields. Without loss of generality, we chose the 0.25° ERA5 reanalysis Hersbach et al. (2020) data as observational data and the state-of-the-art GFDL-ESM4 Dunne et al. (2020) at 1° as our ESM.

### 2.1 Embedding evaluation

Transformations $f, g$ are chosen so that they map observational (OBS) and model (ESM) data to a common embedding space $\mathbf{V^{emb}}$, where all samples are identically distributed. For constructing $f$ and $g$ we need $f(ERA5)$ and $g(GFDL)$ to be unbiased with respect to each other. The transformations need to be chosen such that the embedded data share the same distribution and the same power spectral density (PSD). We assess if they are statistically unbiased towards each other by analyzing their histograms and latitude / longitude profiles, as well as their spatial PSDs (after applying pre-processing transformations). Figure S2 shows that $f(ERA5)$ and $g(GFDL)$ have the same spatial distribution (fig. S2A) with minor differences in temporal statistics shown by the histogram (fig. S2B) and latitudinal/ longitudinal profiles (fig. S2C and fig. S2D).

The individual operations that make up the transformations $f$ and $g$ do not change the large-scale patterns of their respective inputs, as desired for a valid bias correction. The goal of downscaling and bias correction $\omega$ (Fig. 1) is to rely on the unbiased large-scale patterns of the ESM and correct statistics, as well as small-scale patterns. The transformation $g$ conserves the unbiased information from the ESM by construction. Therefore, we want the diffusion model, approximating $f^{-1}$, to also preserve unbiased information.

The extreme case of erasing all detail with large amounts of noise (Fig. 2A) leads to learning the unconditional distribution $p(ERA5)$, which is then not a correction of $GFDL$ but a generative emulation of the ERA5 reanalysis data. We tested this by adding the same amount of noise to the output of our diffusion model that was added to create $g(GFDL)$. This ensures that both the downscaled and bias-corrected fields, as well as the original GFDL fields, lack the small-scale details up to the same point.

To verify that large-scale patterns are preserved by the diffusion model, we compute image similarity metrics between the low pass filtered version of the input of the diffusion model (embedded ERA5 data $f(ERA5)$) and the low pass filtered output of the diffusion model $DM(f(ERA5))$. The output of the low pass filter leaves the large-scale features unchanged. The comparison yields an average structural similarity index (SSIM Wang et al. (2004)) value of 0.8 and a Pearson correlation coefficient of 0.9 for the validation dataset. This verifies that large-scale patterns are well preserved by the diffusion model.

Our diffusion model is able to reconstruct high-resolution fields following the ERA5 distribution from embedded ERA5 fields $f(ERA5)$, with only minor discrepancies in small-scale patterns (fig. S3A).

A comparison between the mean absolute spatial-temporal difference between the first downsampled and then bilinearly upsampled GFDL and ERA5 fields at 0.25 ° yields a mean bias of 0.3 mm/d. The downscaling and bias correction of our diffusion model reduces this bias to 0.1 mm/d (at 0.25°). Our diffusion model thus approximates $f^{-1}$ well, and we successfully created a shared embedding space in which $f(ERA5)$ and $g(GFDL)$ are identically distributed.

### 2.2 Evaluation of downscaling and bias correction performance

We investigate the inference performance of our diffusion model on embedded GFDL data $g(GFDL)$. We compare the downscaling and bias correction performance of our diffusion model to a benchmark consisting of first applying bilinear upsampling followed by QM for bias correction.





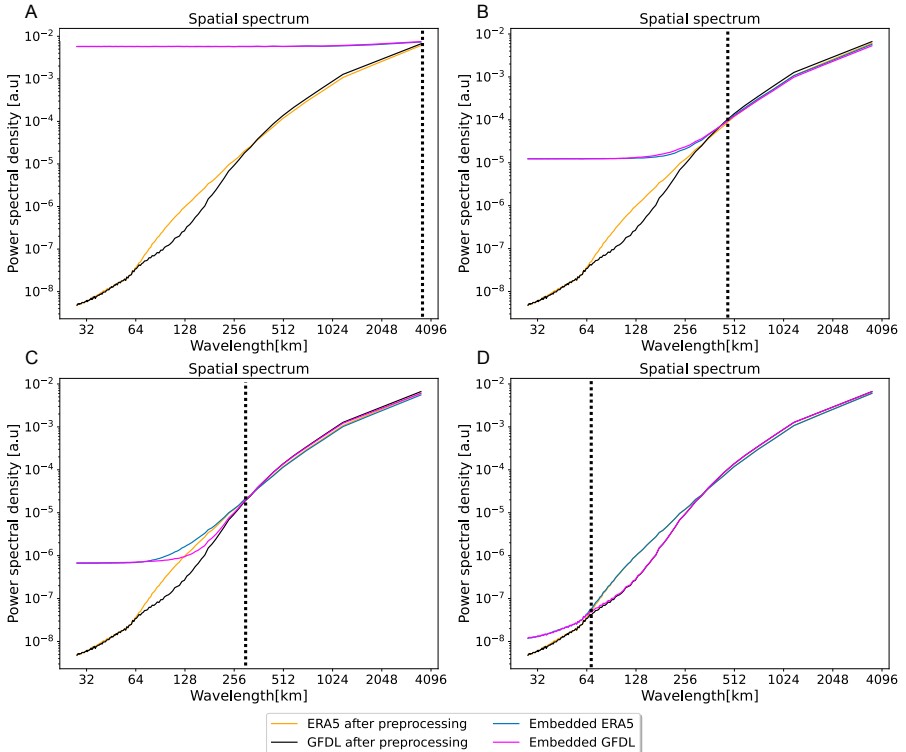

**Fig. 2 Power spectral densities (PSDs) for different choices noising scale of the diffusion model**. The nosing scale $s$ (dashed line) is a hyperparameter that can be chosen depending on the ESM and observational datasets, as well as on the specific task. For the maximal choice of $s$ **(A)** all information in the observations (ERA5) and model simulations (GFDL) is noised and thereby destroyed. Conditioning on pure noise makes the task equivalent to unconditional image generation. The diffusion model will learn to generate observational fields with no relation to the ESM fields. When $s$ is chosen to be minimal, there will be no noising and the conditional generation will directly replicate the condition, i.e. the ESM field. In **(B)** we chose $s$ as the point where the PSDs of the observational and simulated datasets intersect. We then apply sufficiently many forward diffusion noising steps to both datasets, destroying small-scale structure until they agree in the PSD. We call scales smaller than $s$ small scales and scales larger than $s$ large scales. In **(C)** and **(D)**, the effects of choosing a smaller noise scale $s$ are shown. Prior knowledge about the ESM or its accuracy can also guide the choice of $s$.

Figure 3 presents a qualitative comparison between the different individual precipitation fields. The upsampled $GFDL$ fields, as well as our benchmark consisting of the upsampled and QM-corrected $GFDL$ fields, are visually too smooth. They therefore appear blurry compared to the ERA5 precipitation fields despite having the same spatial resolution of 0.25°. Our diffusion model produces high-resolution detailed outputs that are visually indistinguishable from the $ERA5$ reanalysis that we treat as the ground truth. We also compared our diffusion model to a different state-of-the-art diffusion model implementation, EDM Karras et al. (2022). The EDM model was trained for the same number of epochs, while taking twice as long for one. The EDM almost perfectly corrects the spectrum (fig. S4A). However in both the histogram (fig. S4B) as well as in latitudinal and longitudinal profiles (fig. S4C and fig. S4D) the EDM model is far inferior to our proposed diffusion model.

The temporally averaged precipitation fields show that the climatology of the diffusion model-corrected GFDL data (Fig. 4A) and the high-resolution ERA5 data (Fig. 4C) is more accurate and less smooth than the climatology of the GFDL data (Fig. 4B). A comparison between the absolute temporally and absolute spatial-temporally averaged diffusion model corrected GFDL and ERA5 fields (Fig. 4D) yields a bias of 0.29 mm/d. This is a substantial improvement over the original GFDL dataset, which yields a bias of 0.69 mm/d (Fig. 4E). Our diffusion model performs on par



**Fig. 3 Comparative visualization of individual randomly selected samples**. Each row presents four samples of the same dataset. The top row shows GFDL ESM4 data, bi-linearly upsampled to 0.25° to match the other fields. The second row shows QM-corrected and the third row diffusion model-corrected GFDL fields. The bottom row shows samples of the original ERA5 data, which are unpaired to the GFDL fields above. Visual inspection shows that the diffusion model correction greatly improves upon the QM correction in terms of producing realistic spatial patterns, since the QM-corrected fields remain way too blurry compared to the HR ERA5 data. There is no visual difference between the details and sharpness of diffusion model-corrected GFDL fields compared to ERA5.

with the state-of-the-art bias correction performance of our benchmark, which is by design optimal for this task, at 0.26 mm/d (Fig. 4F).

There are large differences between the GFDL and ERA5 data in small-scale patterns (Fig. 5A). The histogram of precipitation intensities (Fig. 5B) also confirms that the ESM is only really accurate



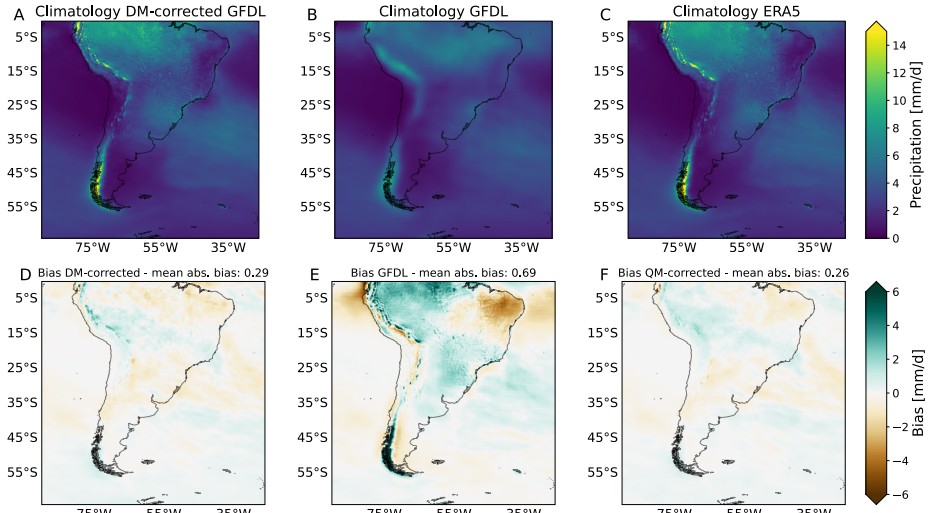

**Fig. 4 Comparison of climatologies and model biases**. The first row shows the climatology of **(A)** the diffusion model-corrected GFDL at 0.25°, **(B)** the GFDL ESM4 model, upsampled to 0.25° and **(C)** the 0.25° ERA5 data. The second row shows the bias of the GFDL and the QM- and diffusion model-corrections, defined as the difference between long-term temporal averages of all validation samples. Specifically, the temporally averaged bias fields with respect to ERA5 are shown for **(D)** the diffusion model correction, **(E)** the uncorrected GFDL and **(F)** the QM correction. Results indicate a substantial improvement of our diffusion model (A) and the benchmark (C) over just upsampling GFDL to 0.25°. The absolute bias on top of each panel is given by the mean absolute value of the differences over the spatial and temporal dimension with respect to ERA5.

for precipitation events up to 40 mm/d, after which the respective frequencies diverge. The latitudinal and longitudinal mean profiles (Fig. 5C and Fig. 5D) indicate the presence of regional biases.

Our framework demonstrates comparable skill to the QM-based benchmark in correcting the latitude and longitude profiles, for which QM is near optimal by construction (Fig. 5C and Fig. 5D). Comparing the histograms (Fig. 5B and fig. S5) shows that our diffusion model is superior compared to the benchmark, strongly outperforming it for extreme values, in particular.

For the spatial patterns and especially the small-scale spatial features, the QM benchmark shows only slight improvements over the original GFDL data (Fig. 5A). The diffusion model is vastly superior in correcting these small-scale spatial patterns (Fig. 5A and Fig. 3) and almost completely removes the small-scale biases, as seen in the spatial PSD.

To verify that large-scale patterns are preserved by the diffusion model, we compute image similarity metrics between the low-pass-filtered embedded GFDL data and the low-pass filtered output of the diffusion model. The comparison yields an average structural similarity index value (SSIM Wang et al. (2004)) of 0.84 and a Pearson correlation coefficient of 0.93, verifying that large-scale patterns are well preserved by the diffusion model.

We also use metrics for precipitation extremes to evaluate the performance of our method. One such metric is the amount of rainfall above a certain percentile. Here we use the R95p metric to investigate the bias correction and downscaling performance of our diffusion model on extreme events. The R95p represents the total annual precipitation from days of heavy rains, calculated as the sum of daily precipitation on wet days (PR > 1 mm/day) that exceed the 95th percentile of our reference period. The difference between the R95p values for the ERA5 and DM corrected GFDL (Fig. S6A), the ERA5 and QM corrected GFDL (Fig. S6B) and ERA5 and GFDL (Fig. S6C), demonstrate that the diffusion model effectively corrects the bias in extreme precipitation events, performing at least as well as the quantile mapping correction. We show that the spatial correlation between the climatologies are improved through our method by computing the Pearson correlation between the temporally averaged fields. The Pearson correlation between ERA5 and GFDL climatology is 0.83,





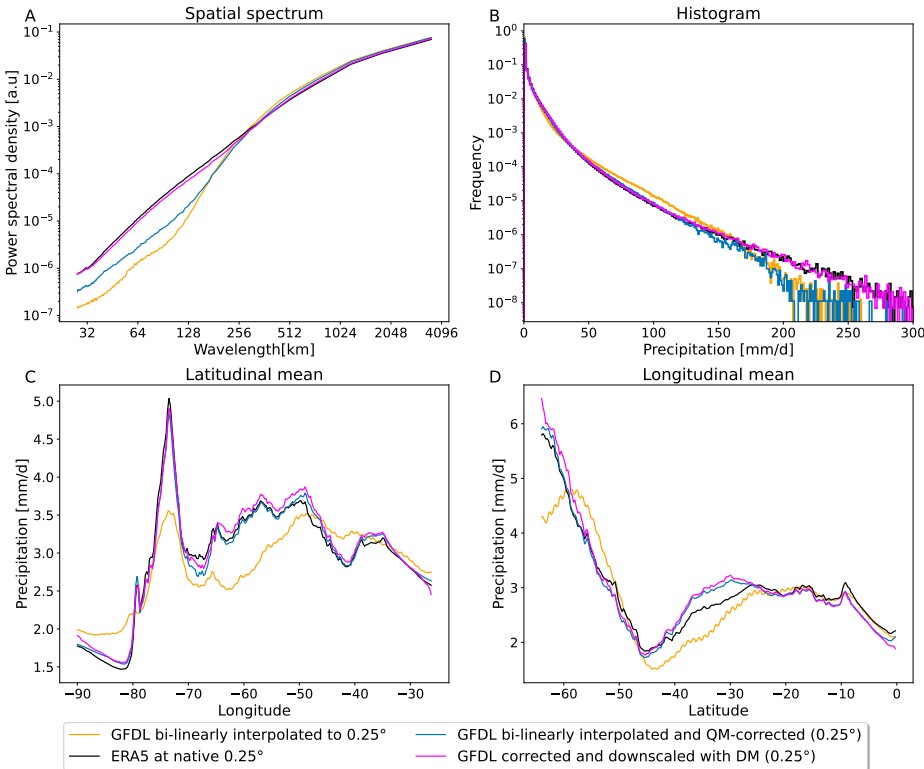

**Fig. 5 Evaluation of our diffusion model's performance for downscaling and bias correction.** Comparison of GFDL (bi-linearly upsampled to 0.25°) (orange) and ERA5 (black) to diffusion model-corrected GFDL (magenta) and QM-corrected GFDL fields (blue) as our benchmark. The Power spectral density (PSD) plot **(A)** shows that the diffusion model corrects the small-scale spatial details far better then our benchmark. The spectrum aligns very well with the high resolution ERA5 target data. The histograms **(B)** as well as the latitude **(C)** and longitude **(D)** profiles show substantial improvements compared to the uncorrected GFDL data.

while the correlation between ERA5 and DM-corrected GFDL is 0.98, which is the same as that for
the QM-corrected GFDL data. We also investigate how our DM captures the statistics of consecutive
dry days (CDD) and consecutive wet days (CWD) compared to the QM benchmark and the raw
GFDL (Fig. S7). Our diffusion model produces superior CDD (Fig. S8A and Fig. S8B) and CWD
(Fig. S8D and Fig. S8E) statistics compared to our QM benchmark and GFDL, as shown in the
difference plots of CDD / CWD.

Our method therefore accurately preserves the large-scale precipitation content, while successfully
correcting small-scale structure of the precipitation fields, as well as statistical biases in terms of
histograms and latitude / longitude profiles (Fig. 5).

## 2.3 Evaluation of ensemble spread

One of the key strengths of our method lies in its capability to generate a diverse ensemble of
downscaled and bias-corrected fields from a single condition. We therefore evaluate the ability of our
diffusion model to represent and produce accurate estimates of uncertainty, a critical aspect for robust
climate modeling and decision-making. We generate a 50-member DM ensemble by running the model
50 times, each conditioned on the same low-resolution ERA5 year, producing one-year trajectories.
The corresponding high-resolution year serves as the ground truth. Our results demonstrate that the
DM ensemble effectively reproduces the correct precipitation patterns, as shown by the close alignment





between the ensemble mean and the high resolution ground truth of ERA5 over the annual cycle (fig. S17). Probabilistic performance, evaluated using CRPS, highlights that the DM significantly outperforms a bilinear baseline, with lower mean CRPS values (0.76 mm/day vs 0.90 mm/day), as well as better temporally and spatially averaged CRPS (fig. S18). Furthermore, we confirm that the DM ensemble produces well-calibrated uncertainty estimates with a spread-skill plot. Our model achieves near-perfect alignment with the 1:1 line, indicating an accurate representation of uncertainty. For more details see Sec. S4.1.

## 2.4 Evaluation on future climate scenarios

Evaluating the performance of downscaling models is crucial for their application in climate impact studies under future climate scenarios. We assess our diffusion model's ability to preserve climate change signals in the underlying ESM simulations by applying it to a high-emission future scenario (SSP5-8.5). Figure 6 compares the relative climate change signal between the late 21st century (2081-2100) and the historical period (1995–2014) for annual mean and annual extreme precipitation. We find that our downscaled 0.25° fields successfully capture the mean precipitation change, closely matching the pattern and magnitude shown in the original 1° GFDL data (Fig. 6A and Fig. 6B). The diffusion model also robustly preserves the climate change signal for extreme precipitation indices, including Rx1Day (wettest day for each year) and R95p (Fig. 6C - Fig. 6F). The spatial patterns of change for the extremes are well-reproduced in the DM-corrected output compared to the original model data. Notably, slight differences are observed in the northwestern domain (Fig. 6C and Fig. 6E), where the DM-correction projects a slightly stronger increase in extreme events under SSP5-8.5. A slight increase in extremes aligns with the diffusion model's bias correction capabilities, reflecting its role in addressing the known under-representation of extreme precipitation in the original GFDL simulations.

Furthermore, we demonstrate that our conditionally trained diffusion model generalizes robustly to unseen future emission scenarios by accurately preserving regional precipitation trends without requiring retraining. We analyze the full annual mean precipitation timeseries from 2015 to 2100 over two representative regions, one exhibiting a strong negative trend and one with a pronounced positive trend (fig. S21). For each region, we compare the annual mean precipitation from the original GFDL SSP5-8.5 data at 1° with the DM-corrected output at 0.25° resolution. The diffusion model consistently preserves the direction and magnitude of the trends found in the original GFDL data across the entire timeseries, for both the negative (fig. S21 blue) and positive trend (fig. S21 red) regions. This demonstrates the model's ability to maintain physically meaningful long-term changes in precipitation, further supporting its generalization capability to future scenarios. Note that the absolute values do not have to coincide, as our model corrects the bias and hence the numerical values. In addition to the trend preservation in the future scenario, we also see that the spatial PSD is corrected by our method (fig. S9A), as well as the histogram (Fig. S9B) and latitudinal/longitudinal profiles (fig. S9C and fig. S9D). Our model can also generalize to unseen climates, preserving the trends, since there is no decrease in performance during inference on GFDL SSP5-8.5 data. Note that our set-up generalized to unseen climate scenarios without any external constraints. The reason why our model preserves trends well is likely given by the fact that the trend is dominated by the large-scale patterns and our model learned to rely on the large-scale patterns of the condition and only generates small-scale patterns.



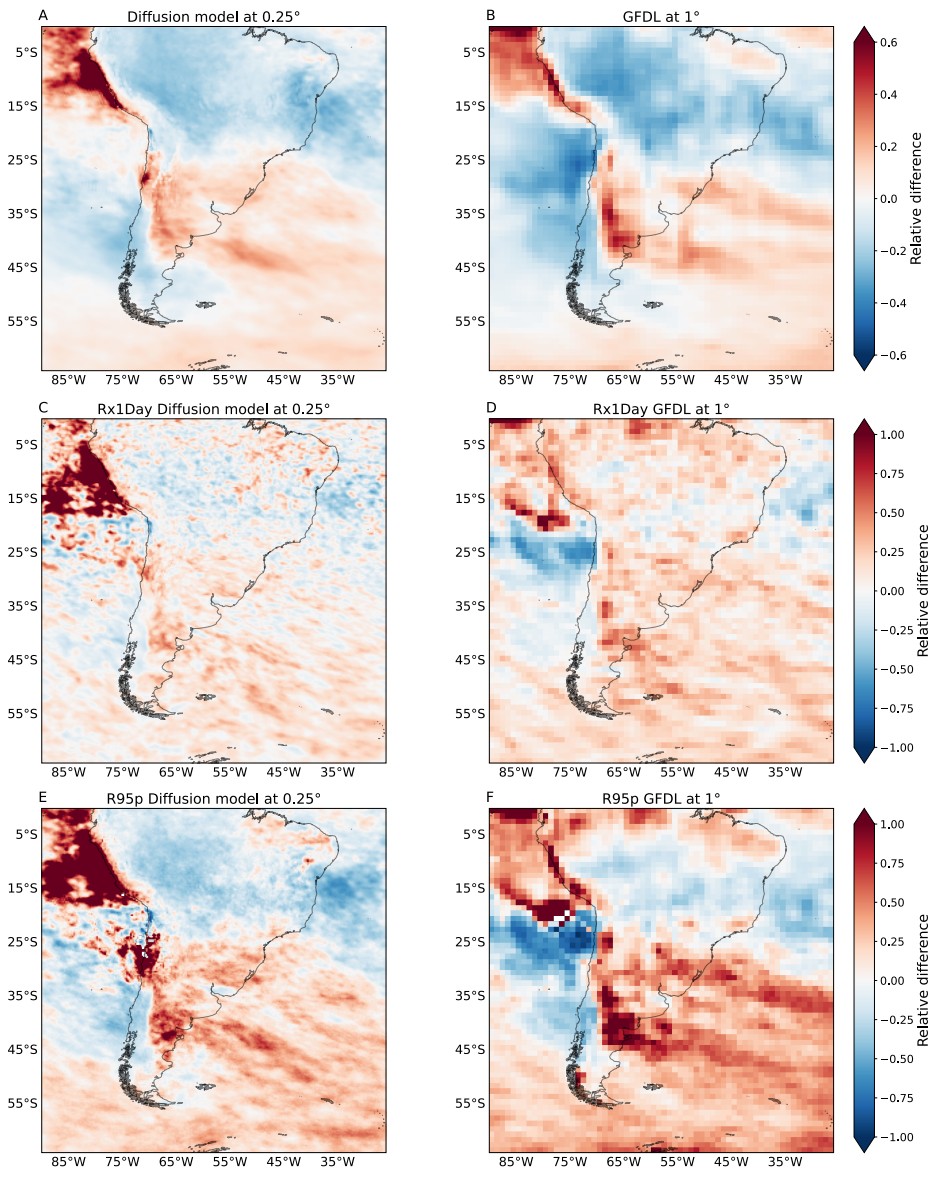

**Fig. 6 Comparison of relative climate change signals**. We compute the relative climate change signal between the late 21st century (2081-2100) under the GFDL SSP5-8.5 scenario and the historical GFDL period (1995–2014). In **(A)** and **(B)**, we show that our diffusion model successfully preserves the mean precipitation climate change signal in the downscaled 0.25° GFDL fields, matching the change of the original 1° GFDL data. Positive values (red) indicate an intensification of precipitation, while negative values (blue) indicate reductions. In **(C-D)** and **(E-F)** we evaluate how well the DM-correction preserves the climate change signal for extreme events in historical and future scenarios. Both the Rx1Day **(C-D)** as well as R95p in **(E-F)** show that the DM-downscaling does preserve the climate change signal for extreme events. There are only slight differences over the north western part of our fields, where the DM-correction predicts more slightly more extremes for the SSP5-8.5 scenario. This is in line with the bias correction capabilities of the DM, correcting the under-representation of extreme precipitation in the original GFDL data.





# 3 Discussion

We introduced a framework based on generative machine learning that enables to do both bias correction and downscaling of Earth system model fields with a single diffusion model. We achieve this by first mapping observational fields and ESM data to a shared embedding space and then applying the learned inverse of the observation embedding transformation to the embedded ESM fields. We learn the inverse transformation with a conditional diffusion model. Although the underlying observational and ESM fields are unpaired, our framework allows for training on paired data (between observations and embedded embedded observations, see above), and therefore any supervised machine learning method can be adopted to the task, which allows for more flexibility. Supervised methods are often superior in performance and more natural for the downscaling application. The diffusion model is trained on individual samples and has successfully learned to reproduce the statistics of observational data. For the observational ground truth, we chose the ERA5 reanalysis, and for the ESM data to be corrected and downscaled, we chose fields from GFDL-ESM4.

Our diffusion model corrects small-scale biases of the ESM fields, while completely preserving the large-scale structures, which is key for impact assessments, especially with regard to extremes and local impacts in terms of floods or landslides. The diffusion model performs particularly well for extreme events where traditional methods struggle. The method improves the temporal precipitation distribution at the grid cell level and surpasses the state-of-the-art approach (quantile mapping) in correcting spatial patterns. The downscaling performance has also been shown to be excellent. The diffusion model manages to generate small-scale details for the low resolution ESM data, that match those of high resolution observations. Our model preserves relevant information form the large scales, such as trends and extremes, and generates bias corrected and downscaled precipitation fields with adequate uncertainties.

We show that our method is robust in the out-of-distribution setting of downscaling and bias-correcting the SSP5-8.5 future emission scenario. It is critical for impact assessments that our model is able to accurately preserve the climate change signal of the original SSP5-8.5 data.

In contrast to Hess et al. Hess et al. (2025), where 3° x 3.75° degree fields are downscaled to 0.75° x 0.9375° resolution, our work focuses on the more challenging task of downscaling from 1° x 1.25° to 0.25° x 0.25° horizontal resolution. This high target resolution is essential to enable impact assessments. At this high resolution, very large amounts of training data would be needed to train an unconditional model. Given the comparably short training data from ERA5, a conditional, rather than an unconditional model as used by Hess et al. (2025), is needed, learning from paired samples. In order to make the training independent of the ESM in this setting, so that the approach can be used for downscaling and bias-correcting any ESM, we developed the strategy as described in detail above and in the methods section. Indeed, comparing results for generated climatologies between our conditional DM and the unconditional consistency model (CM) by Hess et al. Hess et al. (2025), it becomes apparent that the CM struggles to learn the target distribution accurately, leading to substantial blurring (fig. S20) that would hinder applications for impact assessments.

Our method is not specific to ERA5 and GFDL, because the training of the diffusion model does not directly depend on the ESM choice. A specific ESM choice will only modify a hyperparameter in the embedding transformations $f$ and $g$. This, however, requires almost no fine-tuning, as the temporal frequencies can always be matched with quantile mapping. The only parameter that might change for different datasets and use cases is the amount of noise that is added to the observational and ESM datasets. We choose the amount of noise such that the PSDs of the observational ground truth and the ESM fields align beyond a certain scale. This means that we have complete flexibility in deciding which patterns we want to preserve and which we want to correct. This is a major advantage over existing GAN based approaches.

We can decrease the level of detail that is preserved by the diffusion model through increasing the amount of noise added in the transformations $f$ and $g$. This leads to higher uncertainties, and therefore increased variance in the outputs of the diffusion model. It is, however, not guaranteed that this increase in variance corresponds to the uncertainties of the ESM. The amount of noise added is directly proportional to the freedom the diffusion model has in generating diverse outputs and



inversely proportional to the model's ability to preserve large-scale patterns.

The downscaled and bias corrected fields will automatically inherit time consistency between different samples up to the noising scale. This means that ESM fields showing two successive days will still look like two successive days after the correction.

We focused on precipitation data over the South American continent, because of its heavily tailed distribution and the pronounced spatial intermittency. Especially at small scales, precipitation data is extremely challenging to model and therefore serves as a reasonable choice to show the frameworks capabilities in a particularly difficult setting. Regional data is chosen due to computational constraints, yet the diverse terrain of our study region, encompassing land, sea, and a wide range of altitudes, enables robust testing of the downscaling and bias correction performance, also given the substantial biases of the GFDL model in this region. The extension to global scales is straightforward and requires no major changes in the architecture. We intend to include more variables in a consistent manner on a global scale in future research. Optimizing the inference strategy, with speedup techniques such as distillation Luhman and Luhman (2021), to decrease the sampling time will prove helpful in this context. Taking the growing number of diffusion model variants into consideration, a comparison between different approaches can help establish the best method suited for the general task of bias correcting and downscaling ESM fields.

It is straightforward to extend our methodology to downscaling and bias correction of numerical or data-driven weather predictions on short- to medium-range or even seasonal temporal scales. This would not require any fundamental changes to the architecture. This would, however, require a target dataset with sufficiently high resolution. The ability of the diffusion model to not disturb the temporal consistency between samples can be useful in this scenario.





## 4 Materials and Methods

### 4.1 Data

For the study region, we focus on the South American continent and the surrounding oceans. Specifically, the targeted area spans from latitude 0°N to 63°S and from longitude -90°W to -27°E. The training period comprises ERA5 data from 1992-01-01 to 2011-01-01. The range of years included for the evaluation on ERA5 and GFDL spans from 2011-01-02 to 2014-12-01.

### ERA5

ERA5 Hersbach et al. (2020) is a state-of-the-art atmospheric reanalysis dataset provided by the European Center for Medium-Range Weather Forecasting (ECMWF). Reanalysis refers to the process of combining observations from various sources, such as weather stations, satellites, and other instruments, with a numerical weather model to create a continuous and comprehensive representation of the Earth's atmosphere. We use the daily total precipitation data at 0.25° horizontal resolution as the target for the diffusion model.

### GFDL

The climate model output is taken from a state-of-the-art ESM from Phase 6 of the Coupled Model Intercomparison Project (CMIP6), namely GFDL-ESM4 Dunne et al. (2020). The dataset contains daily precipitation data of the first ensemble member (r1i1p1f1) of the historical simulation (esm-hist). The data is available from 1850 to 2014, at 1° latitudinal and 1.25° longitudinal resolution and a daily temporal resolution.

GFDL-ESM4 Dunne et al. (2020) SSP5-8.5 represents a high-emission future pathway. We use daily-resolution data from the CMIP6 archive, provided at 1° latitude and 1.25° longitude spatial resolution, covering the period from 2015 to 2100.

### Benchmark dataset

In order to benchmark our method, we first apply bilinear interpolation to increase the resolution of the GFDL fields from 1° to 0.25°. After that, we apply Quantile delta mapping Cannon et al. (2015) to fit the upsampled GFDL data to the original 0.25° ERA5 data. QM is fitted on past observations and can then be used to correct the statistics of any (past/present) ESM field towards that reference period. We use quantile delta mapping and chose the ERA5 training period from 1992-01-01 to 2011-01-01 as the reference period to fit the GFDL to ERA5. The benchmark dataset to evaluate our approach is then constructed by applying QM to the GFDL validation period (2011-01-02 to 2014-12-31).

### Data pre-processing

The units of the GFDL data are kg m$^{-2}$s$^{-1}$, and for ERA5 m/h. For consistency, both are transformed to mm/d.

Our pre-processing pipeline consists of:

- Only GFDL: rescaling the original 1°× 1.25° GFDL data to 1×1° (64×64 pixel).
- Add +1 mm/d precipitation to each value in order to be able to apply a log-transformation to the data.
- Apply the logarithm with base 10 in order to compress the range of values.
- Standardize the data, i.e. subtract the mean and divide by the standard deviation to facilitate training convergence.
- Transform the data to the range [-1,1] to facilitate the convergence of the training.

As part of the transformation $g$, the 1° GFDL data is bilinearly upsampled. This and the downsampling and upsampling of ERA5 data, which is part of $f$, are already done during preprocessing. The downsampling of 0.25° ERA5 data (256×256 pixel) to 1° (64×64 pixel) is done by only keeping every fourth pixel in each field. For the just mentioned upsampling, we apply bilinear interpolation to increase the resolution from 1° to 0.25°. Note that bilinear interpolation to 0.25° does not increase the amount of information in the images compared to the 1° fields. After preprocessing the data as



described, the embedding transformation $f$ is applied. The diffusion model is trained with the pre-processed $f(ERA5)$ as a condition and the original 0.25° ERA5 data as a target. Before we apply the embedding transformation $g$ we first pre-process the 1° GFDL data by applying quantile delta mapping (QDM Cannon et al. (2015)) with 500 quantiles. The bilinear upsampling is then used to increase the resolution to 0.25×0.25° (256×256 pixels). The preprocessed data are used as input to the embedding transformation $g$. The corresponding output serves as the condition during the inference process of the diffusion model.

## 4.2 Embedding framework

Our framework introduces transformations $f$ & $g$ that map OBS and ESM data to a shared embedding space $f : \mathbf{V^{obs}} \rightarrow \mathbf{V^{emb}}$ and $g : \mathbf{V^{esm}} \rightarrow \mathbf{V^{emb}}$. The goal is to do bias correction and downscaling of ESM fields, i.e., to obtain samples from the conditional distribution $\omega = p(OBS|ESM)$. Training a conditional model to approximate this distribution directly is not possible because OBS and ESM are unpaired. Therefore, we will train the model without the ESM data, only using OBS data and utilize a trick to enable transfer learning and inference on the ESM data. We apply transformations on ESM and OBS such that the resulting datasets are similarly distributed and therefore allow for generalization. The arrows in the diagram of Figure 1 show that we can represent the mapping that achieves the bias correction and downscaling as $\omega = f^{-1} \circ g$. Our idea is to approximate $f^{-1}$ with a neural network $f^{-1} \approx \epsilon$. We chose a conditional diffusion model (DM), denoted by the conditional distribution $p(OBS|f(OBS))$, to approximate $f^{-1} = DM : \mathbf{V^{emb}} \rightarrow \mathbf{V^{obs}}$. The diffusion model (Fig. 1C) is only trained on pairs $(OBS, f(OBS))$. The shared embedding space allows us to evaluate the trained model on ESM embeddings $p(OBS|g(ESM))$, as all embeddings are identically distributed.

### 4.2.1 Constructing the embedding space

The goal of $f$ and $g$ is to map OBS and ESM to a shared embedding space, where $f(OBS)$ and $g(ESM)$ are identically distributed (Fig. 1). To achieve this, both embedded datasets need to be unbiased towards each other. OBS and ESM are biased towards each other in terms of statistical biases between distributions and biases between small-scale patterns visible in the spatial power spectral density (PSD) (fig. S3A).

As mentioned earlier, the input for the embedding transformation $f$ is 0.25° ERA5 data, which is first downsampled, then upsampled and preprocessed. The input to the embedding transformation $g$ is the upsampled and preprocessed 0.25° GFDL data. By first downsampling ERA5 to 1° and then upsampling it to 0.25° we ensure that the fields match the information content of the original 1° GFDL fields.

To remove small-scale pattern bias, we apply a noising procedure analogous to the forward diffusion process as part of $f$ and $g$. Gaussian noise contains all frequencies in equal measure and the Fourier transform of Gaussian noise is itself Gaussian noise, so its power must be equal across all frequencies in expectation. The power spectrum of pure Gaussian noise corresponds to a horizontal line in the spectrum of Fig. 2A, reflecting the fact that it contains all frequencies in equal amounts. Adding noise to an image results in a hinge shape in the PSD of the noisy images (Fig. 2B, 2C and 2D). Increasing the variance of the noise increases its power and, as a result, its PSD will shift upward. Adding noise hence acts as a low-pass filter, while the variance of the added noise determines the cut-off frequency. Increasing variance leads to higher cut-off points as the power of the noisy frequencies increases. Both ERA5 and GFDL data are noised up to the cutoff frequency, denoted by $s$. The scale $s$ is determined by the point where ERA5 and the ESM data (in our case GFDL) start to disagree in their spatial PSDs (Fig. 2), i.e., the intersection between the two. Adding noise in this way ensures that $f(ERA5)$ is unbiased compared to $g(GFDL)$ in the PSD by erasing all information beneath $s$. In our implementation, the transformations $f$ and $g$ utilize the same cosine scheduler as the forward diffusion process to add Gaussian noise to the data. ERA5 data undergoes 50 noise steps within $f$, while $g$ applies the same 50 noise steps to the GFDL data. We ensure that the observational and ESM data have aligned distributions by incorporating Quantile Mapping (QM) directly into the transformation $g$. It is directly applied as pre-processing and only needs to be included in $g$. The quantile-mapped and bilinearly downscaled data is then noised as described above, as part of the embedding transformation. It is important to clarify that QM is not included because the diffusion model is unable to do bias correction. QM is only used as a tool in our framework to ensure that in the embedding space $f(ERA5)$ and $g(GFDL)$ are identically distributed, such that $g(GFDL)$ can be used for the inference of the diffusion model.





### 4.2.2 Determining the noising scale

The choice of the spatial scale $s$ influences up to which scale we correct the spatial PSD. We note that the PSD shows spectral distributions normalized to 1; therefore, we can still observe slight changes above $s$ when small-scale patterns are corrected. The point $s$ is a hyperparameter chosen before training and purely depends on the datasets ESM and OBS and can be adjusted to the specific needs in a given context and task.

In the extreme case, where $s$ is maximal, the conditional images will contain pure noise (Fig. 2A). In this case, the diffusion model is equivalent to an unconditional model. As an unconditional model, the diffusion model will correct all biases at all spatial scales, however, at the expense of completely losing any paring between the condition and the output. We chose $s$ to be at the intersection of the ERA5 and GFDL spectrum around 512 km (Fig. 2B). Thereby, we trust in the ESM's ability to model large-scale structures above the point $s$, which we do not want to correct with the diffusion model.

### 4.3 Network architecture and training

The general architecture of our diffusion model $DM$ consists of a Denoising Diffusion Probabilistic Model (DDPM) architecture Ho et al. (2020) conditioned on low resolution images. For details about diffusion models and conditional diffusion models, see SI Sec. S1.1 and SI Sec. S1.2. This is similar to a single block of a cascading superresolution model. We employ current state-of-the-art techniques to facilitate faster convergence and find the following to be important for convergence and sample quality Saharia et al. (2022b): The memory efficient architecture, "Efficient U-Net", in combination with dynamic clipping and noise conditioning augmentation Ho et al. (2022) turned out to be effective for our relatively small dataset. We adopt the Min-SNR Hang et al. (2023) formulation to weight the loss terms of different timesteps based on the clamped signal-to-noise ratios. The diffusion model architecture utilizes a cosine schedule for noising the target data and a linear schedule for the condition during noise condition augmentation with 100 steps each. The diffusion model is trained to do v-prediction instead of noise prediction. The U-Net follows the $64 \times 64 \rightarrow 256 \times 256$ Efficient U-Net architecture Saharia et al. (2022b). The diffusion model is trained for 100 epochs using the ADAM optimizer Kingma and Ba (2015) with a batch size of 2 and a learning rate of $1e^{-4}$. Note that in the case of fig. S3, where the inference data is also embedded OBS data and there is no ESM data present, the model performs better when being trained and evaluated with 1000 denoising steps, instead of the 100 steps that we used in all our experiments that include ESM data. The model with 100 steps is far superior in training and inference speed and also in correcting the histograms, when correcting ESM data. We also compared the effect of not adding noise in SI Sec. S2.1, a different noise choice in SI Sec. S2.2 during both training and inference, and the effect of not applying QM in SI Sec. S2.3.



## Acknowledgments

### Funding

MA acknowledges funding from the Excellence Strategy of the Federal Government and the Länder through the TUM Innovation Network EarthCare.

SB, and NB acknowledge funding by ClimTip. This is ClimTip contribution #21; the ClimTip project has received funding from the European Union's Horizon Europe research and innovation program under grant agreement No. 101137601.

PH, SB, and NB acknowledge funding by the Volkswagen Foundation.

BP acknowledges funding by the National Key R&D Program of China (2021YFA0718000).

YH acknowledges the Alexander von Humboldt Foundation for the Humboldt Research Fellowship.

## Author contributions

Conceptualization: MA, PB, YH, NB
Methodology: MA, NB, BP, SB
Supervision: NB, SB
Writing—original draft: MA
Writing—review & editing: MA, SB, PH, BP, YH
Investigation: MA
Formal analysis: MA
Software: MA, SB, YH
Data curation: MA
Validation: NB, MA, YH
Funding acquisition: NB
Project administration: NB
Visualization: MA
Resources: YH

## Data and Materials Availability

All data needed to evaluate the conclusions in the paper are present in the paper and/or the Supplementary Materials.

The ERA5 reanalysis data is available for download at the Copernicus Climate Change Service (https://cds.climate.copernicus.eu/cdsapp#!/dataset/reanalysis-era5-single-levels?tab=overview).
The CMIP6 GFDL-ESM4 is available at https://esgf-data.dkrz.de/search/cmip6-dkrz/.

The code will be available on GitHub (https://github.com/aim56009/ESM_cdifffusion_downscaling.git) and Zenodo (https://doi.org/10.5281/zenodo.14849615) Aich (2025).

## Competing interests

The authors declare no competing interests.

## List of Supplementary Materials



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
