# Peer review of "Conditional diffusion models for downscaling & bias correction of Earth system model precipitation"

_EGUsphere, 2025_

## Author Comment (AC3)

This manuscript presents a conditional diffusion model framework for simultaneous bias correction and downscaling of Earth System Model (ESM) precipitation fields. The novelty lies in training the model exclusively on observational data by mapping both ESM and observations into a shared embedding space, where quantile mapping and noise injection help align distributions. The conditional diffusion model then reconstructs small-scale precipitation structures while preserving large-scale ESM patterns. The authors evaluate the method using ERA5 as the observational reference and GFDL-ESM4 as the test ESM, showing improvements over bilinear interpolation with quantile mapping and comparisons with other diffusion approaches. They also highlight strengths in representing extremes, ensemble spread, and future climate scenario preservation.

**Major Issues**

1. The experimental setup is confined to one ESM (GFDL-ESM4) and one reanalysis dataset (ERA5) over a single continental region (South America). While the framework claims generality to "any ESM," the evidence is narrow. Without testing multiple models or regions, it is unclear whether the embedding and conditional framework is robust to diverse ESM biases and precipitation regimes. Furthermore, the noise-scale hyperparameter, chosen at the spectral intersection, is dataset-specific and may require fine-tuning across contexts, raising concerns about general applicability.

We thank the reviewer for this important comment. Indeed, our experimental validation focused on a single ESM and region. We have framed this as a proof-of-concept to demonstrate the viability of our proposed method in a challenging environment.

We originally chose the South American region, because it presents a diverse and complex test case for our method. It includes complex orography given by the Andes that produce highly variable (elevation-dependent) precipitation, alongside significant land-ocean interfaces that govern coastal weather patterns. The region contains diverse precipitation regimes from very dry regions like the hyper-arid Atacama desert to the deep convection of the Amazon rainforest.

However, we completely agree with the reviewer's suggestion and in order to strengthen our claims, we performed additional experiments with I.) a different region and II.) a different ESM.

I.) We chose a region of the same spatial extent, from 0.75°N to 64.5°N latitude and 42°E to 105.75°E longitude, over South Asia. The method of preprocessing the new data remains the same as for South America. The high-resolution ERA5 target data consists of 256x256 grid cells at a 0.25°x0.25° resolution, while the low-resolution ESM data is initially projected to 64x64 pixels at a 1°x1° resolution. We trained a diffusion model with the same objective as with our South America dataset. The DM maps from the embedded ERA5 data to the high-resolution ERA5 target data. At inference, we use the DM to bias-correct and downscale the GFDL inference data (over the same timespan as in the South America case) by mapping it to the shared embedding space (one that is specific to the South Asia data)

and then correcting it with the diffusion model. The result is shown in **Figure S22**, demonstrating that our approach works equally well compared to South America:

The power spectrum of the DM-corrected data matches that of the ERA5 target data, while the first bilinearly downscaled and then quantile-delta-mapping-corrected GFDL data lacks variability in the small spatial scales. QDM is only a minor improvement for the small scale variability compared to the original GFDL data that was only bilinearly downscaled (fig. S22A). In the histogram, both the DM and QDM corrected data improve especially the less extreme precipitation, for example in the range 10 mm/d - 30 mm/d, compared to the original GFDL data. The DM slightly outperforms QDM, in particular up to extremes of around 90 mm/d (fig. S22B). In terms of latitudinal and longitudinal mean, both the DM and QDM improve the large biases of the original GFDL data and show a similar distance to the ground truth ERA5 data, while the DM is even a slight improvement over the quantile mapped data (fig. S22C & fig. S22D).

Fig. S22 Evaluation of the diffusion model's downscaling and bias correction performance over South Asia. Comparison of precipitation fields from the DM-corrected GFDL model (magenta) with the original GFDL output (orange), the quantile mapping benchmark (blue), and ERA5 (black) as the reference. All datasets are at a 0.25° resolution. The original GFDL data and the data before QM were bi-linearly upsampled. (A) The Power Spectral Density (PSD) plot shows the DM's superior correction of small-scale features. (B) The histogram of daily precipitation shows that both DM and QM corrections improve the distribution of moderate events (e.g., 10–40 mm/d) compared to the raw GFDL data. (C, D) The latitude and longitude mean profiles show that both correction methods offer similar improvements over the original GFDL data.

II.) We repeated our experiments with the MPI-ESM-HR model, without re-training the diffusion model. We fixed the experimental setup and only switched the dataset. We first projected the 0.9375°×0.9375° resolution ESM data to a 1°x1° resolution with CDO. The power spectra of GFDL and MPI-ESM start to disagree at around the same point in the PSD plot. This means the training hyperparameters are the same for GFDL and MPI-ESM. Therefore, there is no need to train the DM again, which maps embedded ERA5 to high-resolution ERA5 data. At inference, we use the DM to produce bias-corrected and downscaled MPI data, given the embedded MPI data. As part of the embedding transformation, we also applied QDM for the MPI data.

The result is shown in Figure S23. In the power spectrum, we see large improvements in the DM-corrected data over the quantile delta-mapped benchmark data as well as the raw bilinearly downscaled MPI data (Figure S23A). In the histogram, we see that the DM-corrected data outperforms the benchmark and the bilinearly upsampled MPI data (Figure S23B). The latitudinal and longitudinal means show a similar distance to the ERA5 data for both the DM and the benchmark. Both significantly improve upon the MPI baseline (Figure S23).

Fig. S23 Evaluation of the diffusion model's downscaling and bias correction performance for a different ESM. Comparison of precipitation fields from the DM-corrected MPI model (magenta) with the original MPI output (orange), the quantile mapping benchmark MPI (blue), and ERA5 (black) as the reference. All datasets are at a 0.25° resolution. The original MPI data and the data before QM were bi-linearly upsampled. (A) The Power Spectral Density plot shows that the DM almost perfectly restores spatial variability across all scales, significantly outperforming both the raw MPI and the benchmark at small scales. (B) The histogram of daily precipitation demonstrates the DM's superior ability to reproduce the frequency of extreme events, matching the ERA5 reference far better than the benchmark. (C, D) The latitudinal and longitudinal mean profiles show a similar correction of our DM and the benchmark over the raw MPI data.

Comment Part II: "Furthermore, the noise-scale hyperparameter, chosen at the spectral intersection, is dataset-specific and may require fine-tuning across contexts, raising concerns about general applicability."

The reviewer is correct that our method is not completely plug-and-play for any arbitrary ESM without retraining. Our claim of generality applies to the methodological framework itself, not to the trained diffusion model that we present, although the above example shows that the trained DM can be successfully applied to both GFDL-ESM4 and MPI-ESM without additional training. Our framework requires the diffusion model to be retrained when using a different target region or target dataset. The learned diffusion model's denoising process is specific to that region's climatology.

The noise-scale hyperparameter *s* is intentionally designed to be user-specific, and indeed, the DM needs to be re-trained for different choices. The choice of s determines the spatial features that will be preserved by our diffusion model. Without preserving any features, the model would be a pure reanalysis emulator with no relation to the underlying ESM data that we want to correct. The ability to set a cutoff, which can be chosen according to our heuristic (i.e. where the spatial PSD of the ESM output and observations cross) or based on prior knowledge, is a crucial contribution of our method. We proposed to choose the point where ESM and ERA5 start to disagree in the PSD as s. This provides a clear guidance for users' choices and ensures the model correctly preserves the large scales for which the ESM is trusted while correcting the small scales for which it is biased. However, the model training only depends on the ESM/region via the one hyperparameter s, and it is therefore still very flexible and requires minimal adjustment. Different ESM fields with the same resolution will presumably have biases of a similar magnitude. Therefore, to correct multiple ESMs at once, one can use the most heavily biased model to select the hyperparameter and then train a single diffusion model to correct all ESMs at inference to save computational resources. This will thus correct all ESMs at the same scale, so that the output can be directly compared. We have revised the introduction and discussion to clarify the role and data dependence of the noising scale s to avoid confusion (lines 370-377).

We have revised the manuscript (abstract, introduction, result), to further contextualize our generality claims and describe the experiments (lines: 21-22, 156, 165-167, 279-295). We discussed the results in (lines 357-364) to give a fair overview of the generalization performance of our framework, which still may require retraining in some scenarios.

2. The choice of benchmark—bilinear upsampling followed by quantile mapping (QM)—is somewhat too weak given recent literature. QM is indeed the statistical baseline, but the field has seen GAN-based approaches (cycleGANs, conditional GANs), CNN-based super-resolution, and unconditional consistency models that have been applied to similar downscaling tasks. Although the authors briefly compare with Hess et al. (2025) and an EDM model, the evaluation remains limited and not systematic. A stronger study would include comparisons against multiple state-of-the-art baselines (GAN, VAE, transformer- or CNN-based super-resolution methods) under consistent experimental conditions.

We thank the reviewer for this comment and agree that a more comprehensive comparison against a broader set of state-of-the-art baselines strengthens the claims of our manuscript. To address this, we trained two additional state-of-the-art machine learning models as suggested by the reviewer, a CNN and a transformer-based one, and evaluated them in the same experimental setup.

**The new baselines are:**

- CNN-based: We utilize the same UNet architecture that serves as the backbone of our diffusion model. The model is trained as a deterministic, end-to-end model with the same task as the diffusion model. The UNet is trained to map embedded ERA5 fields to high-resolution ERA5 fields with the MSE loss.
- 2. **Transformer-based:** To evaluate the capability of transformer-based models, we augmented our UNet architecture by incorporating self-attention blocks. Following modern practices, we integrated the self-attention layers into the bottleneck and at several feature resolutions within the encoder and decoder paths. It was trained with the same objective as the CNN baseline.

At inference, we apply the diffusion, CNN-based, and transformer-based models, respectively, to correct the same embedded GFDL data. We compare them using our main evaluation metrics in **Figure 24 & lines 231-239**.

We updated the manuscript to include the results from these new experiments in **Figure 24**. to present a side-by-side comparison using our key evaluation metrics. The most striking difference is in terms of PSD, where the diffusion model (DM) significantly outperforms the two additional benchmarks by generating more realistic small scale patterns that align much better with those of the high-resolution ERA5 fields **(Fig. S24A)**. The histogram shows that all three methods correct the data well, so they align with the ERA5 distribution **(Fig. S24B)**. Also in terms of latitudinal and longitudinal mean all methods perform similarly **(Fig. S24C & Fig. S24D)**.

Crucially, both the CNN and transformer-based models are deterministic and hence do not allow for uncertainty estimates or ensemble predictions. In contrast, the stochastic diffusion model naturally reflects the one-to-many characteristic of downscaling from lower to higher resolution. This is a fundamental advantage of the diffusion architecture over both. The ability to model small-scale structures very well with a stochastic model shows that the DM is preferable to CNN and transformer-based models for our problem setting.

We would like to highlight that we already compare our method with a state-of-the-art VAE model, a VQ-VAE model that serves as another generative baseline in addition to Hess et al. (2025) [6] and the EDM model.

Thank you for also suggesting comparisons with traditional generative models such as conditional GANs or cycleGANs. However, we believe these models are no longer state-of-the-art in image synthesis and come with significant drawbacks compared to diffusion models, particularly when applied to climate data like precipitation (see details below). Considering the extensive time required to fine-tune and train these models, we argue that their inclusion would not be the most meaningful addition to this study for the following reasons.

Diffusion models have been regarded as the state-of-the-art in image generation for several years [2,3,4]. While GANs have shown effectiveness in tasks such as natural image super-resolution, recent advances in diffusion models have surpassed them in terms of image synthesis quality. Diffusion models achieve superior results on various benchmarks while offering improved stability and addressing fundamental issues such as mode collapse [2]. Dhariwal and Nichol (2021) [2] demonstrate that diffusion models consistently outperform GANs (and VQ-VAE) in generating high-quality images, as verified by metrics like FID, IS, Precision, and Recall.

The limitations inherent to GANs are especially pronounced for non-natural image data, such as precipitation climate data. While GANs can perform well in natural image generation by trading off diversity for fidelity to produce high-quality samples, this trade-off often results in incomplete coverage of the target distribution. This shortfall can lead to critical aspects of the distribution being missed, particularly for precipitation extremes, which are a key focus of our modeling. Additionally, GANs are prone to mode collapse, where certain parts of the distribution (e.g., rare or extreme precipitation events) are underrepresented [2].

The adversarial training process of GANs is another major drawback, often suffering from instability, difficulty in convergence, and requiring careful, resource-intensive tuning of the generator-discriminator balance. We have encountered these challenges firsthand during the training of a CycleGAN [5]. CycleGAN also does not fit the training setup that requires a conditional GAN.

Recently popular super-resolution GAN-based methods like SR-GAN also rely heavily on perceptual loss, derived from a pretrained VGG model, to ensure perceptual similarity in natural image tasks. However, this approach is unsuitable for precipitation downscaling. VGG is trained on natural images and does not capture the physical or statistical characteristics of precipitation fields. In our task, perceptual similarity has no meaningful correspondence with the statistical accuracy required for high-resolution precipitation data. Our primary goal is to accurately represent the statistical properties and extremes of precipitation, which are critical for climate modeling, rather than achieving visual similarity. The use of perceptual loss could impose artificial constraints on the generation process, resulting in outputs that appear visually consistent but fail to reflect the true physical characteristics of precipitation fields.

Unlike traditional super-resolution tasks, our study addresses both **super-resolution** and the **denoising** of low-resolution (LR) embedded precipitation fields. Diffusion models inherently offer better control through their iterative generation process, which allows for fine-grained adjustments at each step. This iterative process, combined with the denoising objective, makes diffusion models particularly robust in handling noisy conditions. Singh et al. (2023) [1] highlight that GANs can struggle with denoising tasks, amplifying artifacts or failing to reconstruct fine details. The ability to handle noisy data during the downscaling process without any issues represents a significant advantage of diffusion models for our application.

By explicitly modeling the entire conditional distribution p(HR|LR), our approach enables robust uncertainty quantification, a critical feature for climate modeling. In contrast, GANs only learn to sample from the conditional distribution, which introduces the risk of mode collapse.

In summary, while GANs have been widely used for natural image super-resolution, they are not designed to meet the unique requirements of our task, which include:

- Handling Noisy Data
- Modeling Rare Events
- Uncertainty Quantification
- Stability and Robustness
- Capturing Statistical Properties
- Physical Consistency over perceptual loss

Given these challenges, we argue that revisiting comparisons with GAN-based models is not the most meaningful addition to our study, as their limitations for precipitation downscaling and modeling are well-documented and DMs are arguably better suited for the challenges of our task. We are confident that diffusion models offer significant advantages in terms of stability, statistical accuracy, and uncertainty quantification.

We argue that our method outperforms the state-of-the-art methods we used for comparison, namely VQ-VAE, consistency models, the state-of-the-art EDM diffusion model, and now, a transformer-based and a CNN-based super-resolution model. We are confident that by comparing our method against these strong baselines, we now provide an extensive comparison with other state-of-the-art machine learning methods.

3. The manuscript does not clearly articulate how the proposed model differs from existing diffusion-based downscaling and bias correction efforts. For example, Wan et al. (2024) combined diffusion with optimal transport, while EDM (Karras et al., 2022) provides another diffusion benchmark. The authors claim advantages in efficiency and data efficiency, but the conceptual distinction between their conditional embedding approach and these prior diffusion frameworks is not fully elaborated. Is the main novelty the embedding trick with QM + noise to align distributions? Or is it the conditional supervision on observational embeddings? This needs further discussion.

We thank the reviewer for this suggestion. We understand that a more detailed discussion of our model's conceptual novelty in the context of existing diffusion-based methods is necessary. We have revised the manuscript to explicitly elaborate on these distinctions.

To address the reviewer's specific point, our main novelty is that we introduce a framework that proposes a proper training objective and alleviates the constraint of needing paired data in order to bias-correct and downscale climate model data with ML methods. To use the vast majority of state-of-the-art ML methods (such as Diffusion Models, transformers, VAEs, CNNs, and cGANs), we need to ensure that the training and inference distributions are identically distributed and we need to choose a proper training objective (defined by what is the input and output data and what is the task the model is trained for). For the bias-correction and downscaling task that we are interested in, the source data comes from

an ESM while the target data is a reanalysis product. Training an ML model on this data comes with two key issues:

**Issue I:** In terms of the long-term climate simulation, there is no correspondence between the observed and simulated meteorological fields for a certain day. As a result, the source data (low-resolution ESM output) and target data (high-resolution observational/reanalysis data) are inherently unpaired. A specific day in a climate model simulation does not correspond to the same calendar day in the observational dataset. This lack of pairs prevents a direct supervised mapping from a low-resolution climate model data to a high-resolution reanalysis.

**Issue II**: The second option is to train a model only on reanalysis data and at inference apply it to climate model data. This approach, however, is only justified if the training and inference distribution are independent and identically distributed (i.i.d.). The systematic biases in the ESM data compared to reanalysis data violate this constraint, leading to poor generalization and unreliable outputs.

So to answer the question directly, the key component of our approach is the introduction of the embedding space, which I) removes the need for paired ESM-observation training data and II) creates a match train-inference distribution, thereby allowing us to train a bias-correction and downscaling model. Without this technique, a machine learning model trained on ERA5 data and applied to GFDL data will not generalize due to a shift between the training and inference distributions. Introducing the embedding space also provides a clear training objective for the ML model. By first mapping both the ERA5 as well as the climate model data to a shared embedding space (with only statistical transformations), we construct a setting where both the training and inference data follow the same distribution.

We can then train a ML model to map the embedded ERA5 data to the clean ERA5 data. At inference we can then apply the trained model to the embedded GFDL dataset and map it to the "bias free", high-resolution ERA5 distribution while preserving spatial patterns of the underlying GFDL data. Being able to set a cutoff, which can be chosen automatically or based on prior knowledge, is an important contribution of our method. Previous methods like cycleGAN [5] do not offer this flexibility. This cutoff allows us to explicitly preserve spatial scales above the cutoff scale and correct features below the cutoff scale. This is crucial because the definition of downscaling requires that some properties or structures are preserved. Without preserving any features, the model would be a pure reanalysis emulator with no relation to the underlying ESM data that we want to correct.

The framework allows us to perform ablation studies with different models, such as CNNs, transformers, VAEs, and consistency models, enabling us to plug in the latest and most suitable machine learning model for the task. This ensures the model correctly preserves the large scales for which the ESM is trusted, while correcting the small scales for which it is heavily biased.

Differences to the above mentioned EDM paper and Wan et. al:

- We view EDM not as a competing application (competing with our method), but as a
  fundamental framework for designing state of the art diffusion models. The
  innovations in EDM, like preconditioning, noise scheduling, and sampler design are
  orthogonal to our contribution, as they can be employed to improve the performance
  of diffusion models in general.
- Wan et al. (2024): Wan et al. combine a diffusion model for downscaling with an Optimal Transport (OT) model for bias correction. OT learns a non-parametric map between distributions, which is notoriously computationally expensive and complex to optimize, especially for high-dimensional climate fields. Our QM-based embedding is a highly efficient statistical transformation, making our framework significantly more scalable and tractable for large-scale, real-world datasets. This directly supports our claims of improved efficiency. Our work also additionally provides a comprehensive analysis of our model's performance on real-world climate data, demonstrating its ability to correct biases, downscale accurately, and capture extremes, uncertainties, and trends. This stands in contrast to Wan et al. (2024), who evaluated their method on idealized systems, namely the one-dimensional Kuramoto-Sivashinsky equation and the Navier-Stokes equation with Kolmogorov forcing. Our focus on validating the framework on operational observational and ESM fields represents a critical step towards practical application. We test the behaviour in out of distribution cases like extreme events and future scenarios as well as ensemble performance and thoroughly evaluate the resulting data.

We updated the introduction (lines 143-158) to better highlight that our main novelty is the introduction of the embedding framework that establishes clear training and inference objectives and overcomes the necessity of paired data for machine learning-based bias correction and downscaling. Furthermore, we demonstrate that generative diffusion models perform excellently at correcting biases, downscaling accurately, while capturing extremes, uncertainties, and trends.

4. While the results on extremes (R95p, Rx1Day) and SSP5-8.5 trends are promising, the metrics are limited. Extreme event validation could be broadened with tail-focused skill scores, quantile-specific errors, or return-level analyses. For future scenarios, the manuscript shows preservation of mean and trend, but it remains unclear whether the method could distort physical consistency (e.g., covariance with other variables, conservation constraints). Since diffusion models are inherently stochastic, an evaluation of physical realism constraints would be useful.

We thank the reviewer for his suggestions. We agree that extensive evaluation of extreme events is important for investigating the robustness of our framework. We agree that the initial metrics for extremes could be extended, however we already want to highlight that we already evaluate on a broad range of metrics (see list of bullet points below). To address this, we performed a new analysis according to the reviewer's suggestion, focused on the return periods of heavy precipitation events, which directly relates to the tail of the distribution.

We additionally performed a return-level analysis over a 20-year period (1995-2014). To ensure the robustness of this validation, we analyzed the return periods for two distinct high-precipitation thresholds, a moderately extreme event (>50 mm/day) and a very extreme event (>80 mm/day). We compared ERA5 as a reference to the GFDL data (bi-linearly upsampled to 0.25°) and our DM-corrected GFDL data.

Our analysis shows that the raw GFDL model exhibits a significant wet bias, substantially underestimating the return periods for both 50 mm/day events (3.33 years) and 80 mm/day events (4.60 years) compared to the ERA5 reference (4.11 and 7.38 years respectively). Our DM-corrected model successfully corrects this, resulting in more realistic average return periods of 4.18 and 7.98 years. While the raw GCM produces an unrealistic, spatially very diffuse pattern of frequent extremes, our model learns to reproduce the sharper, and locally much more constrained heavy rainfall extremes that are characteristic for the ERA5 reference data (see Fig. S25). As the improvements are consistent across both thresholds, this indicates that our framework robustly corrects the tail of the GFDL precipitation distribution, also supporting the physical plausibility of the downscaled fields.

Fig. S25 Empirical return periods of extreme precipitation events. The figure compares the spatial distribution of return periods (in years) for moderately extreme (50 mm/day, top row) and very extreme (80 mm/day, bottom row) events from our DM-corrected GFDL (left), the ERA5 reference (middle), and the raw GFDL model (right). The return period for each grid cell is calculated as the inverse of the mean annual frequency of exceedances. The raw GFDL model shows a significant wet bias, with unrealistically short return periods (spatially averaged values are shown above each plot) and diffuse spatial patterns. Our DM mitigates this bias, resulting in sharper, more localized patterns and average return periods that closely match the ERA5 reference for both thresholds.

We have added our analysis to the results section (lines 278-284). We believe this substantially strengthens our conclusions regarding the model's performance for extreme events.

This additional tail-focused metric is designed to complement the primary analyses we conduct in this study. Our existing validation already included:

- The full precipitation histogram (Fig. 5B), which showed our model's ability to match the ERA5 data even for high-intensity events in the tail of the distribution.
- We conducted an evaluation of both wet extremes like R95p (Fig. S6), Rx1Day (Fig. 6) and consecutive wet periods CWD (Fig. S7 and Fig. S8), as well as consecutive dry periods CDD (Fig. S7 and Fig. S8), demonstrating our model's skill across a diverse range of extreme event characteristics. Our evaluation covers not only high-intensity rainfall but also the persistence of both wet and dry conditions.
- We show that our model preserves the climate change signals for Rx1Day and R95p under the SSP5-8.5 scenario, confirming its generalization capabilities for future scenarios.
- As a response to another reviewer, we now also added scatter plots for CDD and CWD for a longer validation period (see response to comment 4)

Part II of the comment: "For future scenarios, the manuscript shows preservation of mean and trend, but it remains unclear whether the method could distort physical consistency (e.g., covariance with other variables, conservation constraints). Since diffusion models are inherently stochastic, an evaluation of physical realism constraints would be useful."

We agree that evaluating the physical consistency of generative models is important. However, the scope of our current study presents some inherent limitations on the types of constraints that can be applied.

As you correctly point out, assessing the preservation of covariance with other variables is a key test of physical realism. Our current framework, however, is **univariate**, focusing solely on the downscaling and bias correction of precipitation. Since other physical variables (like temperature, pressure, or wind) are not inputs or outputs of the machine learning model, we cannot evaluate the inter-variable covariance structure. We successfully built a robust univariate model and we regard extending the framework to more complex, multivariate systems as future work.

In a single-variable context for a variable like precipitation, defining strict physical constraints (e.g., mass or energy conservation) is non-trivial. The "physical realism" we aim for is primarily demonstrated through the generation of spatially coherent, plausible precipitation fields and the accurate reproduction of the full statistical distribution of the observational data in our work. By showing that our model preserves the mean state, long-term trends, and key statistics of extremes, we provide strong evidence that it operates within a physically plausible regime. Note that in Section 2.1, we also report that the low pass filtered version of the input and output of the DM have a high similarity. This shows that the large-scale spatial patterns are well preserved by the DM and therefore the fields inherit large-scale physical realism from the ESM.

To address your comment directly in the manuscript, we have added a paragraph to the Discussion section (lines X-Y), where we explicitly state that the evaluation of multivariate physical consistency is a critical next step for future research and necessary if one wants to extend the method to multiple variables, in for example an operational setting.

5. A central claim is that the proposed method is independent of the chosen ESM because the diffusion model is trained only on observations. However, in practice, the embedding transformation g requires quantile mapping of ESMs, which is itself model-dependent. Thus, some degree of ESM-specific adjustment is unavoidable. The manuscript should acknowledge this limitation and discuss how sensitive results are to the chosen reference period, quantile mapping scheme, and observational dataset.

We thank the reviewer for highlighting this. We rephrased our initial claim of being "independent of the chosen ESM" to avoid being misinterpreted. The reviewer is correct that quantile mapping itself is always dependent on the chosen ESM. Our intention was to highlight that our core component, which demands most of the computational budget, i.e. the diffusion model, is trained exclusively on observations, but we acknowledge that the overall workflow is not completely model agnostic.

As already remarked in comment 1, it is correct that training of the diffusion model depends on the hyperparameter *s*, for which a choice can be made in dependence of the ESM, namely by checking where the spatial PSDs for ESM fields and observations cross. The bias, and therefore the scale up to which we trust the ESM, could differ between models, requiring an adjustment of *s*.

We also acknowledge (see response to comment 1) that the choice of the hyperparameter *s* (the spatial frequency cutoff) introduces an implicit ESM-dependency. Ideally, *s* would be tuned for each ESM based on its specific spatial bias characteristics or the degree to which we trust that ESM.

In practice, however, a goal could be to efficiently correct multiple ESMs. Training a diffusion model for each ESM is computationally infeasible. Our framework solves this by allowing a single diffusion model to be trained and then applied to the whole ensemble of ESMs. This is achieved by selecting a conservative value for s, typically determined by the ESM in the ensemble with the lowest spatial variability details. This will also ensure comparability between different ESMs.

This choice represents a deliberate trade-off; we can sacrifice optimization for each ESM in exchange for a highly efficient and scalable workflow. The primary benefit is that the computationally intensive training is performed only once, creating a single correction model applicable to an entire ensemble of ESMs. We contend that this is a justifiable compromise for achieving a broadly applicable and computationally efficient method.

Ultimately, the choice of *s* determines the scale of large-scale structures preserved from the ESM. For an ensemble of state-of-the-art models with comparable resolutions, we expect

the range of optimal *s* values to be relatively narrow. Therefore, the practical impact of selecting a single conservative value is likely modest, especially when weighed against the substantial gain in computational efficiency that allows a single model to correct an entire ensemble.

We have clarified these points in the manuscript and adjusted the wording to be more specific (lines 377-388).

To answer the dependence of the method on the quantile mapping scheme, the reference period and the target dataset:

- The reviewer correctly notes that the final outputs depend on the quantile mapping scheme. In this work, we deliberately chose Quantile Delta Mapping (QDM), which is critical, as it allows the application of our method to climate change projections. Simpler quantile mapping methods are known to improperly remove or weaken the climate change signal (trend) from the ESM. QDM is a state of the art trend preserving bias correction method, which ensures that the long term trends in the future scenario ESM data is preserved. While other trend preserving methods exist, QDM is a robust and widely used approach and we chose it for its computational efficiency. We included a statement in the paper, acknowledging the dependence on the specific quantile mapping scheme (lines 387-388).
- We acknowledge the reviewer's point on the sensitivity to the chosen reference period. First of all, a strength of our method is that it is very performant while being only trained on a relatively small amount of data (1992-2011). We view this as a strength, as it could be used in a low data scenario where observations are only available for a few years (lines 438-442). ERA5 data from the satellite era is available from 1980 onwards. A limitation of any machine learning model is its reliance on the statistical properties of the training data. A reference period of a few decades may not fully capture the complete range of climatic variability and in particular very extreme events. What makes our model robust to this issue is that we always rely on the large scale features of the ESM and heavy extreme events will be dominated by the larger scales. Our model can reproduce the events on the large scales (which it is conditioned on) and does not have to predict them based on its internal representation of the training data (which would lack numerous heavy extremes). Our model is data efficient because it only needs to learn how the small scale patterns behave, given the large scale patterns. However, the generative model can still only learn to represent small scale features it has seen during training and heavy extreme events might not be captured perfectly when moving too much out of the training distribution. Therefore, when data efficiency plays no role it is best to choose the largest available period for training the model. In our study, we opted to show that the model is very data efficient because it only generates the small scale variability, given the large scale one (see Fig. 27).

Overall, the performance of machine learning models on extreme climate events is inherently limited by the representativeness of the training data. Our model has some advantages over unconditional methods that need to learn the full data distribution, because it inherits large scale patterns from the ESM. We have added a sentence in the manuscript to acknowledge this limitation (lines 460-462).

 Finally, we completely agree that the choice of the observational target dataset (in our case ERA5) is fundamental to the output. Our diffusion model is trained to learn the statistical distribution of the target data and therefore the results are completely dependent on the dataset by design. We selected ERA5 as our target dataset due to its high resolution, accuracy, and widespread use.

**Recommendation**

The manuscript introduces a promising and technically creative approach that leverages conditional diffusion for a challenging problem in climate modeling. However, the current version has limitations in experimental breadth, benchmark rigor, and clarity of novelty relative to existing diffusion approaches. I recommend major revision before publication. The authors should expand the benchmark comparison, better articulate how their method diverges from and improves upon existing diffusion-based methods, and provide more robust multi-model/multi-region evaluations to strengthen the claim of general applicability.

We thank the reviewer again for the constructive suggestions. We believe that we have addressed the concerns by adding new experiments with a different ESM and region to demonstrate generality, and we included stronger CNN and Transformer benchmarks to validate our model's performance. The revised text also clarifies that our core novelty is the embedding framework itself, which solves the critical problems of unpaired data and distribution shift.

- [1] V. Singh *et al.*, "Performance Analysis of GANs for De-Noising Images," *2023 International Conference on Information Technologies (InfoTech)*, Varna, Bulgaria, 2023, pp. 1-7, doi: 10.1109/InfoTech58664.2023.10266875.
- keywords: {Training;Visualization;PSNR;Image coding;Surveillance;Noise reduction;Generative adversarial networks;GAN;CNN;Neural Network;Residual blocks;RELU;PSNR;MSE},
- [2] Dhariwal, Prafulla, and Alexander Nichol. "Diffusion models beat gans on image synthesis." *Advances in neural information processing systems* 34 (2021): 8780-8794.
- [3] Rombach, Robin, et al. "High-resolution image synthesis with latent diffusion models." Proceedings of the IEEE/CVF conference on computer vision and pattern recognition. 2022.
- [4] Saharia, Chitwan, et al. "Photorealistic text-to-image diffusion models with deep language understanding." *Advances in neural information processing systems* 35 (2022): 36479-36494.
- [5] Hess, Philipp, et al. "Physically constrained generative adversarial networks for improving precipitation fields from Earth system models." *Nature Machine Intelligence* 4.10 (2022): 828-839.
- [6] Hess, Philipp, et al. "Fast, scale-adaptive and uncertainty-aware downscaling of Earth system model fields with generative machine learning." *Nature Machine Intelligence* (2025): 1-11.

**Anonymous Referee #2**

This manuscript proposes a framework to downscale GFDL-ESM4 using a conditional diffusion model trained only on ERA5. The authors align the train/test distributions by (i) applying quantile mapping (QM) to the ESM data to remove large-scale biases and (ii) adding carefully chosen noise so that both ERA5 and GFDL are projected into a shared embedding on which the conditional diffusion model is trained and applied.

**What I like**

- Focusing on precipitation is well motivated; it remains one of the hardest fields for ML downscaling and bias correction.
- Within the scope of their data, the authors conduct a relatively deep analysis and explore key hyperparameters. The SI is useful for additional insights
- The idea to select a noise (cutoff) scale via the PSD relationship between ERA5 and GFDL seems new; the model then matches small-scale (high-wavenumber) PSD to ERA5 while preserving large-scale ESM information. This design appears effective based on PSDs, trend preservation, and overall fidelity.

**Clarify the embedding vs. preprocessing story**

- Early in the paper I read the approach as latent-space manipulation after mapping both datasets into a shared space. Later it became clear that the shared embedding is achieved via preprocessing (QM + controlled noising), and the diffusion model learns to reverse the noising conditioned on the preserved large scales.
- This sequencing is a bit confusing and contributes to statements about "no dependency on the test dataset" and "no ESM OBS pairing" being misread.
- Concretely: this is supervised training on ERA5 and inference on preprocessed GFDL that has been mapped into the same embedding. I recommend making that pipeline explicit with a schematic and a sentence like we preprocess ERA5 and GFDL to a shared embedding (via QM + noising up to scale s); we train on ERA5 in this embedding and apply the learned conditional reverse process to embedded GFDL at inference.

Reference line in the paper: "We map observational and ESM data to a shared embedding space, where both are unbiased towards each other and train a conditional diffusion model to reverse the mapping."

We thank the reviewer for this helpful comment. We understand that our use of the term "embedding" can be misinterpreted as a learned latent space, while it is in fact referring to part of our preprocessing pipeline.

Your summary of our method is accurate:

- I.) We define a preprocessing pipeline for the ESM, as well as for the observational (ERA5 in our case) data. After preprocessing, we refer to the resulting datasets as 'embedded ERA5' and 'embedded ESM' data. Both now share similar statistics.
- II.) The diffusion model (DM) is trained to map the embedded ERA5 data to the clean ground truth ERA5 data, which is hence in principle independent of the ESM, but dependent on the hyperparameter *s* that sets the spatial scale below which the DM corrects biases.
- III.) At inference we use the pre-trained DM to map the preprocessed ESM(GFDL-ESM4 and additionally MPI-ESM in the revised manuscript) data to the ground truth ERA5 data. During training, the DM did not have to change large scale information during training as those are not affected by the noise preprocessing. The model only had to generate small scale structures on (small) scales that were covered by noise. Therefore the model will preserve large-scale spatial features by construction.

We have adjusted the line cited by the reviewer, but also updated the structure of our explanation of the embedding procedure in the introduction (lines 122-158) to avoid potential confusion.

**Quantile mapping and potential leakage**

- Please specify exactly how and when QM is fit and applied, if QM parameters are estimated using years that later appear in validation, or worse, from the future period, there is a risk of data leakage and trend distortion.

We thank the reviewer for this comment. Quantile Delta Mapping (QDM) was calibrated using a historical period from 1992-01-01 to 2011-01-01. The QDM was fitted using ERA5 (1992-2011) as the reference and GFDL (1992-2011) as the model dataset. The mapping was then applied to the GFDL data from the non-overlapping validation period 2011-01-01 to 2014-12-01.

To obtain 20 years of DM-corrected GFDL data (1995–2014), we conducted an experiment where we used this period from ERA5 as the reference and for fitting and applying the QDM. For historical data, the ground truth ERA5 data always exists, so when the goal is to produce the best possible bias correction, it seems justified to use the same historical period for fitting and applying the QDM. Trend distortion does not seem to play a significant role in this case. For future scenarios, we fully agree with the reviewer's argument regarding trend distortion.

For the future scenario, we use the 1995 to 2014 period of ERA5 as reference data and the historical GFDL data as the model input to fit the QDM. We then apply this mapping to the full time period of the GFDL SSP5-8.5 data (2015–2100). To address the reviewer's concern, no data leakage or trend distortion could occur for the future period in this QDM setup. We now clarify the exact periods chosen for QDM in the paper (lines 511-515).

**Scope: broaden temporal and regional tests**

- The analysis is relatively deep but narrow in scope.
- Extend to at least one additional region with different regimes.

We completely agree with the reviewer's suggestion that testing our model on an additional region would strengthen the paper's claims. In response to this comment and a similar one from Reviewer I (comment 1), we have performed a new set of experiments on a region over South Asia (0.75°N to 64.5°N latitude and 42°E to 105.75°E longitude), which has a very different climatology compared to South America.

The results confirm the robustness of our method. Our diffusion model (DM) successfully corrects systematic biases and generates realistic small-scale spatial variability in the new region, performing on par with QDM regarding histograms and significantly improving over it in terms of reproducing the power spectrum.

These new results are presented in detail in the supplementary material (**Figure S.X**). To avoid repetition, a full description of the experimental setup is provided in our response to Reviewer I (comment 1). We are confident that this new analysis demonstrates the general applicability of our framework beyond the initial domain.

- Use a longer temporal validation, including seasonality coupled with temporal behavior (autocorrelations, wet/dry spell durations, event persistence), and spatial/temporal scatter comparisons between train (ERA5) and test (GFDL-embedded) across the full-time span.

We thank the reviewer for his constructive feedback and the suggestion to expand upon the temporal validation of our model. This is a critical aspect, and we agree that a more detailed analysis strengthens the paper. We have performed a series of new experiments on a longer validation timescale (1995-2014) to specifically address the reviewers suggestions.

First, we want to clarify our model's design regarding temporal consistency. Our diffusion model is trained to downscale individual timesteps separately. Consequently, the day-to-day temporal evolution of large-scale climate features is directly inherited from the driving low-resolution GFDL input. The model's task is not to learn temporal dynamics, but to add physically plausible, high-resolution spatial details conditioned on the large-scale state of each day.

As the reviewer correctly implies, the temporal consistency of these newly generated small-scale features is not explicitly enforced by the training (but nevertheless emerges as we show below). To address the reviewer's comment, we also mention in the discussion that we see a video diffusion model that processes time series instead of single snapshots to fully guarantee temporal consistency all the way, as future work (lines 428-430). The following experiments were designed to validate that the resulting high-resolution time series exhibits realistic temporal behavior.

**1. Temporal autocorrelation**

To address the reviewer's comment, we now compute the temporal autocorrelation for the full 20-year time series, to check if the model produces temporally realistic, persistent weather patterns rather than independent noise.

As shown in **Figure S26**, the autocorrelation of the DM-corrected output (orange) very closely matches the ERA5 reference data (blue). The decay of correlation over a 5-day lag is nearly identical, indicating that our model preserves the natural persistence of precipitation events. It is a notable improvement over the bi-linearly interpolated GFDL data, which exhibits too much persistence.

**2. Seasonal Event Persistence and spatial scatter (CDD/CWD)**

The reviewer suggested analyzing "spatial/temporal scatter comparisons between train (ERA5) and test (GFDL-embedded)". We did not fully understand this because train ERA5 is in a different dataspace than the embedded-GFDL data, but we believe the most insightful approach in this context is to evaluate the spatial patterns of key temporal metrics, such as spell durations as also suggested by the reviewer. This addresses seasonality and event persistence simultaneously. The motivation for this analysis is to confirm that our diffusion model not only generates the correct global mean statistics but also correctly distributes these temporal characteristics in space.

We conducted new experiments, computing Consecutive Dry Days (CDD) and Consecutive Wet Days (CWD) for every grid point over the 20-year period for two distinct seasons: June-July-August (JJA) and December-January-February (DJF). **Figure SX** compares the per pixel CDD & CWD durations of the bi-linearly upsampled GFDL data and the DM-corrected GFDL data (at 0.25°) against the ERA5 reference.

Consecutive Dry Days (CDD): The raw GFDL data (green points) exhibits a strong dry bias, systematically overestimating the length of dry spells compared to ERA5. The points are significantly scattered. In contrast, our DM-corrected output (blue points) shows a clear improvement. The points are clustered more tightly around the 1:1 line, indicating that our model corrects the dry bias and more accurately reproduces the spatial pattern of drought persistence in both JJA and DJF.

**Consecutive Wet Days (CWD):** The raw GFDL data (green) is again highly scattered and fails to capture the correct distribution, particularly underestimating long wet spells. The DM-corrected output (blue) brings the CWD climatology much closer to the ERA5 reference, with a significantly reduced bias for the longer wet spells.

This directly provides evidence for the ability of our DM to handle event persistence.

We would also like to point out that aspects of this longer-term validation were already present in our original manuscript (see Fig. 6) and we already looked at CDD and CWD (Figs. S7, S8).

Finally,we would like to highlight that we also show our main evaluation metrics in **Figure SX** to confirm that the model performs comparably on this longer evaluation period. The DM performs similarly well when evaluated on 1995-2014 **(Fig. SX)** and for 2011-2014 (Fig. 5).

Fig. S28 Model performance evaluation over the extended 1995-2014 period. This figure replicates the analysis from Fig. 5 but for the longer evaluation period of 1995-2014 to demonstrate the robustness of the results. It compares our DM-corrected GFDL (magenta), the QM benchmark (cyan), and the bi-linearly interpolated GFDL (orange) with the ERA5 reference (black), all at 0.25° resolution. The findings are consistent with those from the shorter 2011-2014 period. (A) The Power Spectral Density plot confirms the DM's superior ability to correct small-scale spatial patterns.(B) The histogram shows that the DM more accurately reproduces the frequency distribution of precipitation events than the benchmark. (C, D) In the mean profiles, both the DM and the benchmark offer comparable improvements over the raw GFDL data.

We added all three plots for the additional experiments to the SI (Fig. X,Y,Z) to strengthen our claims in the paper, regarding the temporal consistency. We discuss the results in the SI lines(181-191). We also expand upon how future work could investigate video diffusion models to process full time series instead of single frames to guarantee temporal consistency as well as focus on evaluating it (lines 441-443).

**Choice of the cutoff scale s**

- You present s as the PSD-based crossover that yields strong performance. That is reasonable.
- Compare to alternative mechanisms (e.g., providing a noise channel explicitly to a strong CNN baseline, or conditioning variants in diffusion that target high-frequency losses).

We thank the reviewer for this interesting comment, we appreciate the suggestions. We agree that there are different ways to choose the cutoff scale s. The most suitable approach could in practice depend on the specific application. The core role of the cutoff scale s is to define which spatial scales from the ESM should be preserved in the bias-corrected and downscaled output of the DM, and which should be corrected. There is no strict mathematical requirement for the exact scale, and our heuristic of choosing the PSD intersection is a guick, simple, and effective way.

One has to differentiate the way the noise scale *s* is determined and how it is integrated into the model. Our approach of setting a specific noising scale is through noising the DM input, and our way of choosing the noising scale is through the PSD intersection. We understand the reviewer's suggestion as options for the way the noising scale is integrated into the model, by either adding the noise as an additional channel or through a specific loss. In both cases, the amount of noise has to be known before training and has to be determined explicitly. We acknowledge that both alternatives are plausible and might be useful in certain scenarios. Both, however, come at certain additional costs.

A CNN with an additional noising channel would, first of all, increase the input dimensionality by adding the noise channel alongside the ESM data. The model would then be tasked not only with bias correction and downscaling but also with learning an additional task. The model needs to learn that the ESM input can be trusted only on large spatial scales but not on small scales and it would need to learn automatically which information to trust and which to discard. This additional task is far from trivial and would take away model capacity, potentially reducing the model's efficiency for its primary downscaling task. Our method simplifies this by filtering out the unreliable information before the model is applied.

A sophisticated loss function could, in principle, also achieve a similar result. However, designing such a loss is not trivial. A loss focused only on high-frequencies could not enforce our primary requirement that the large-scale patterns of the ESM must be preserved. For that, we would need a more complex loss consisting of a low- and a high-frequency loss. This approach is more engineered and introduces a weighting factor for the high- and low-frequency losses. This introduces an additional ESM dependent hyperparameter that would likely need to be tuned for different datasets.

We argue that our approach is more efficient and conceptually easier to implement. However, there might be use cases where other mechanisms for including the noise might be suitable. Generally, our proposed framework also supports other noise conditioning mechanisms.

**Baselines and diversity**

- The test set lacks model diversity (single ESM), and the baselines are limited.
- Add at least one more ESM with different small-scale biases.

We thank the reviewer for this constructive feedback and understand that testing our method on another ESM is important for demonstrating its robustness.

ALso in response to this comment and a similar point raised by Reviewer I (Comment 1), we have conducted a new experiment using a different earth system model, MPI-ESM-HR.

The results confirm the robustness and generality of our framework. Our DM successfully corrects the biases in the MPI-ESM-HR output, significantly outperforming the QDM baseline in restoring small-scale spatial variability and improving the precipitation distribution. Importantly, our pre-trained DM could be applied directly to this new ESM without any retraining, highlighting an advantage of our approach.

These new results are presented in the supplementary material (**Figure S.Z**). To avoid repetition, a full description of the experimental setup and the results is provided in our response to Reviewer I (Comment 1). We are confident that this analysis demonstrates our framework is not tailored for a single ESM and can be applied more generally.

- Add strong ML baselines (e.g., diffusion/SR variants trained on down/upsampled ERA5 pairs, competitive CNN/Transformer SR models) alongside standard statistical methods.

We thank the reviewer for this important suggestion. We agree that comparing our method against other strong machine learning baselines is essential to contextualize its performance.

In response to this comment and a similar one from Reviewer I (Comment 2), we have now trained and evaluated two additional state-of-the-art models as requested: a CNN-based and a transformer-based super-resolution model.

The results show that while all advanced methods (our DM, CNN, Transformer) produce similar histograms, latitudinal and longitudinal means compared to ERA5, our DM is superior in generating realistic small-scale spatial variability, as shown by the power spectral density (see Fig. X). Furthermore, as a generative model, our DM offers the crucial advantage of producing ensembles for uncertainty quantification, a feature that the deterministic CNN and transformer models lack.

This comprehensive comparison has been added to the SI in **Figure SY**. To avoid repetition, a full description of the model architectures and training setup is provided in our detailed response to Reviewer I (Comment 2). We are confident that these additions provide a robust comparison against relevant state-of-the-art methods and clearly highlight the advantages of our proposed approach.

**Transformations and ablations**

- Data undergo heavy transformations (log, scaling, etc.). Please include ablations on these choices and demonstrate their effects.

We thank the reviewer for this comment. We agree that understanding the impact of preprocessing is an important aspect of building robust machine learning models. Our data preprocessing pipeline includes adding +1 to the data, followed by a log-transformation and then standardization (subtracting the mean and dividing by the standard deviation), and finally a projection to the range [-1, 1]. All these operations are based on well-established practices in both the machine learning and precipitation modeling communities.

Normalizing input features is a standard practice when training deep neural networks. Standardization ensures that all input variables are on a comparable scale (e.g.[1]). This prevents variables with large magnitudes from dominating the learning process and helps maintain smooth gradients, leading to faster convergence.

State-of-the-art weather and forecasting models, like GraphCast [2] or Aurora [3], also apply standardization by subtracting the mean and dividing by the standard deviation. Feature scaling, in our case to the range [-1, 1], is also a common preprocessing technique (used in [4,5]) that can empirically help to get faster convergence of the gradient descent algorithm.

Precipitation is a difficult variable to model due to its highly skewed distribution. It is characterized by a large number of zero or near-zero values and long tails of rare, extreme events. Applying a log transformation in combination with adding +1 in order to compute the logarithm for precipitation values of 0 is a way to improve the learning process for an ML model (for example also used in [4,5]). The transformation compresses the range of the data and makes it closer to a Gaussian distribution. This allows for a smoother landscape for gradient-based optimization. It is common practice for processing precipitation.

The reviewer suggests an ablation study to empirically demonstrate the effect of these transformations. An ablation study would involve retraining our entire model from scratch for different combinations of preprocessing. Given that our preprocessing is considered common practice in machine learning and, in case of the log transformation for precipitation data, the expected outcome of such an ablation would likely result in training instabilities and a potential degradation. At best, we would not expect any significant improvement. Experiments comparing different preprocessing strategies are computationally very expensive and would in our view not yield new scientific insight relevant to the scope of our study.

- [1] Goodfellow, Ian, et al. *Deep learning*. Vol. 1. No. 2. Cambridge: MIT press, 2016.
- [2] Lam, Remi *et al.*,Learning skillful medium-range global weather forecasting.*Science***382**,1416-1421(2023).DOI:10.1126/science.adi2336
- [3] Bodnar, C., Bruinsma, W.P., Lucic, A. *et al.* A foundation model for the Earth system. *Nature* **641**, 1180–1187 (2025). https://doi.org/10.1038/s41586-025-09005-y
- [4] Hess, Philipp, et al. "Deep Learning for Bias-Correcting CMIP6-Class Earth System Models." *Earth's Future* 11.10 (2023): e2023EF004002.

[5] Hess, Philipp, et al. "Physically constrained generative adversarial networks for improving precipitation fields from Earth system models." *Nature Machine Intelligence* 4.10 (2022): 828-839.

**Figures**

- Use a good and uniform color map and clearer legend/contrast for Figure 3.

We have plotted the fields using a logarithmic scale and a different colorbar, to improve the clarity and contrast of Figure 3.

- In sample 4 (upper-left corner), precipitation patterns appear to change; please comment on whether this is intended regeneration of small-scale structure consistent with large scales, or an artifact.

Firstly, we assume that the reviewer is referring to sample 4 from Figure 3. We are not entirely sure which specific change the reviewer refers to, as all individual fields in Figure 3 show different (changing) fields. In general, for this figure, we do not expect the last row (ERA5) to visually match any of the other fields we show.

The only fields where similarity matters are those of GFDL (row I) with QM-corrected GFDL (row II) and DM-corrected GFDL (row III), but **only** for the same sample index. Also, note that the QM-corrected GFDL and DM-corrected GFDL fields are only expected to agree on their large-scale spatial structures. For that, please also see the analysis done in **lines 268-272** that shows that the large-scale spatial structures are preserved well, but are still not entirely identical between GFDL and DM-corrected GFDL. Changing the small scale patterns can also, to a small degree, change larger-scale structures (our experiments and overall evaluation throughout the paper show in a good way). The point of this figure is to show how, for different samples, the GFDL, QM-corrected, and DM-corrected fields mostly agree with each other on large-scale spatial patterns. We also want to show how the DM additionally improves spatial variability, therefore showing a resolution similar to ERA5.

When analyzing sample 4 specifically, we see that the QM-corrected field is visually similar to the GFDL field. The large-scale features of the DM-corrected field also look similar to those of the QM-corrected field. We assume the reviewer is referring to a specific, slightly extreme precipitation event that appears (slightly) red in the QM-corrected field but is less intense in the DM-corrected one.

For any generative AI method, especially a stochastic one, it is difficult to interpret the model's "intent" for a single, specific output. Our analysis showed a strong overall agreement on large scales, but this agreement is not expected to be perfect in every detail. We believe the observed behavior is within our expectations, as the DM's function is to correct biases by changing smaller-scale patterns. In this case, it has weakened precipitation over some areas

and strengthened it in others to align better with the overall ERA5 rainfall patterns. We do not see any contradictions with any of our other metrics.

**Inputs, reproducibility, and generalizability**

- What are the input channels to the model (precip only, or multivariate conditioning such as humidity, winds, temperature)? Please list them explicitly.

We do not mention any other variables in the paper, because the only variable we use in this study is precipitation.

- Provide exact config files/scripts (including how s is computed from two PSDs) to ensure full reproducibility.

We chose *s* visually at the point where the RAPSD of the ESM and ERA5 start to disagree, we have a dedicated notebook (paper\_fig\_eval\_embedding\_trafo.ipynb) where we compare the statistics of embedded ERA5 and embedded GFDL for PSD, histogram and latitudinal & longitudinal profile. We added a comment to the code to highlight that the effect of different noising scales can be compared and observed there. One can use our framework with different diffusion models, like EDM, and of course also with a variety of noise schedulers (different ones are used in state-of-the-art models). Different choices there will lead to different numerical values of *s*. We therefore created this notebook to test different models / noise schedulers and investigate the value for *s* empirically.

- Clearly mention that new ESM will require training the model again due to precprocessing - it seems to be missing the in the text.

We thank the reviewer for this remark. We added this in the discussion (Lines 378-392).

I recommend major revision. The approach is promising and potentially impactful, but the current version requires broader validation, clearer methodological framing, and stronger baselines. I would be happy to review a revised manuscript if the authors choose to resubmit.

We thank the reviewer again for the constructive comments. We believe that the additions and revisions we made to the manuscript address the reviewer's concerns.